# RNA promotes phase separation of glycolysis enzymes into yeast G bodies in hypoxia

Gregory G Fuller[1], Ting Han[2], Mallory A Freeberg[1†], James J Moresco[3‡], Amirhossein Ghanbari Niaki[4], Nathan P Roach[1], John R Yates III[3], Sua Myong[4], John K Kim[1]*

[1]Department of Biology, Johns Hopkins University, Baltimore, United States; [2]National Institute of Biological Sciences, Beijing, China; [3]Department of Chemical Physiology, The Scripps Research Institute, La Jolla, United States; [4]Department of Biophysics, Johns Hopkins University, Baltimore, United States

*For correspondence:
jnkim@jhu.edu

Present address: [†]EMBL-EBI, Wellcome Genome Campus, Hinxton, United Kingdom; [‡]Center for the Genetics of Host Defense, UT Southwestern Medical Center, Dallas, United States

Competing interests: The authors declare that no competing interests exist.

**Abstract** In hypoxic stress conditions, glycolysis enzymes assemble into singular cytoplasmic granules called glycolytic (G) bodies. G body formation in yeast correlates with increased glucose consumption and cell survival. However, the physical properties and organizing principles that define G body formation are unclear. We demonstrate that glycolysis enzymes are non-canonical RNA binding proteins, sharing many common mRNA substrates that are also integral constituents of G bodies. Targeting nonspecific endoribonucleases to G bodies reveals that RNA nucleates G body formation and maintains its structural integrity. Consistent with a phase separation mechanism of biogenesis, recruitment of glycolysis enzymes to G bodies relies on multivalent homotypic and heterotypic interactions. Furthermore, G bodies fuse in vivo and are largely insensitive to 1,6-hexanediol, consistent with a hydrogel-like composition. Taken together, our results elucidate the biophysical nature of G bodies and demonstrate that RNA nucleates phase separation of the glycolysis machinery in response to hypoxic stress.

## Introduction

Cells perform many diverse activities that are spatially and temporally organized into non-membrane-bound compartments that often form transiently and can display solid, gel, or liquid-like properties. Liquid-like structures formed through phase separation of protein polymers display fast internal rearrangements, undergo fusion and fission, and exchange components with the surrounding solvent (*Alberti et al., 2019*; *Hyman et al., 2014*).

Recently, we and others demonstrated that glycolysis enzymes coalesce into membraneless cytoplasmic granules called glycolytic bodies (G bodies) in hypoxic stress in yeast, *C. elegans*, and mammalian cells (*Jang et al., 2016*; *Jin et al., 2017*; *Miura et al., 2013*). In yeast, the presence of G bodies correlates with accelerated glucose consumption, and impairing G body formation leads to the accumulation of upstream glycolytic metabolites (*Jin et al., 2017*). These data suggest that during hypoxic stress, when oxidative phosphorylation is inhibited, G bodies form to enhance the rate of glycolysis by concentrating glycolysis enzymes. In *C. elegans*, hypoxia rapidly induces the formation of foci containing glycolysis enzymes near presynaptic release sites in neurons (*Jang et al., 2016*). A phosphofructokinase mutant incapable of punctate localization disrupts synaptic vesicle clustering in neurons, suggesting that coalescence of glycolysis enzymes promotes synaptic function (*Jang et al., 2016*). However, the mechanism of G body formation remains poorly understood.

G bodies have a number of features common to known phase-separated bodies. In addition to being non-membrane bound, some G body components, including phosphofructokinase 2 (Pfk2),

contain intrinsically disordered regions (IDRs), a feature of proteins that undergo phase transitions (*Jin et al., 2017*). The IDR is required for Pfk2 localization to G bodies (*Jin et al., 2017*). In addition, G body formation in mammalian cells is inhibited by addition of RNase to the culture media, suggesting that RNA is required for G body integrity (*Jin et al., 2017*).

Many other phase-separated structures contain RNA. For instance, stress granules contain mRNAs stalled in translation initiation (*Buchan et al., 2008*) and involve protein-protein and IDR interactions between mRNA binding proteins (*Panas et al., 2016*; *Protter et al., 2018*). Nucleoli, which are sites of ribosomal RNA processing, also form by phase separation (*Brangwynne et al., 2011*; *Protter et al., 2018*). For some proteins, RNAs can promote phase separation in vitro (*Elbaum-Garfinkle et al., 2015*; *Zhang et al., 2015*) or even phase separate by themselves (*Jain and Vale, 2017*; *Van Treeck et al., 2018*). Furthermore, RNase treatment can disrupt mature granules such as P bodies, demonstrating the importance of RNAs in the structural integrity of RNP granules (*Teixeira et al., 2005*). The identity of bound RNAs can be important for phase separation. For example, the yeast Whi3 protein phase separates in the presence of its substrate RNA, *CLN3*, but not in the presence of total RNA (*Zhang et al., 2015*). Additionally, RNAs that phase separate alone from yeast total RNA in vitro are enriched in stress granules, suggesting that RNA may drive phase separation of stress granule proteins in vivo (*Van Treeck et al., 2018*).

In contrast, other RNA binding proteins display the opposite behavior. For Pab1, a core stress granule component, high concentrations of RNA prevent phase separation in vitro (*Riback et al., 2017*). Microinjection of RNase A can induce aggregation of nuclear FUS protein in vivo, suggesting that, in this case, high levels of RNA oppose phase separation. RNA binding-defective mutants of TDP-43 display an increased propensity to phase separate in vitro and in vivo and addition of TDP-43 RNA substrates allows TDP-43 to remain soluble (*Maharana et al., 2018*; *Mann et al., 2019*).

Taken together, reconciling these disparate behaviors in which some RNAs promote formation of stress granules and P bodies, while others antagonize phase separation of proteins such as TDP-43 and FUS will require targeted in vivo approaches to determine how mature granules are affected by RNA. In granules for which RNA facilitates RNP granule formation, RNA may function by forming a scaffold to promote multivalent interactions (*Fay and Anderson, 2018*; *Jain and Vale, 2017*; *Langdon and Gladfelter, 2018*). Interestingly, metabolic enzymes, including some involved in glycolysis, bind mRNAs (*Beckmann et al., 2015*; *Matia-González et al., 2015*). However, the physiological roles of this RNA binding remain poorly understood.

In this study, we show that G bodies are novel RNP granules formed through a phase separation mechanism in vivo. We identify the common mRNA substrates of the G body resident glycolysis machinery and demonstrate the essential role that RNA plays in G body biogenesis and maintenance in vivo. Thus, our data suggest a model where, in response to hypoxic stress, when cellular demand for energy must be met solely through glycolysis, G bodies form through multivalent protein-protein and protein-RNA interactions to enhance the rate of glycolysis.

## Results

### Analysis of the RNA-binding proteome uncovers core G body constituents

To identify the RNA binding proteome in *S. cerevisae*, we incorporated 4-thiouridine (4SU) into RNA in log phase yeast cells grown under normoxic conditions and combined photoactivatable-ribonucleoside-enhanced cross-linking (PAR-CL) with oligo(dT) affinity purification and tandem mass spectrometry (PAR-CL-MS, *Figure 1A*) as previously described (*Baltz et al., 2012*; *Castello et al., 2012*). Stringent washes and separation by SDS-PAGE removed non-crosslinked proteins. We identified 259 mRNA-binding proteins (mRBPs) as defined by having >2 counts in +UV mass spec and 0 counts in −UV mass spec or an FDR < 10% (see Materials and methods). Nearly half of these mRBPs were non-canonical RBPs that do not contain conventional RNA binding domains. Many of these have been identified as mRBPs in yeast and other organisms (*Beckmann et al., 2015*; *Matia-González et al., 2015*), thus validating our approach (*Figure 1B*). In addition, our study identified 69 novel mRBPs in yeast (*Figure 1B*, *Supplementary file 1*). Consistent with previous studies, we identified 10 glycolysis enzymes as mRBPs and other studies identified five additional glycolysis enzymes

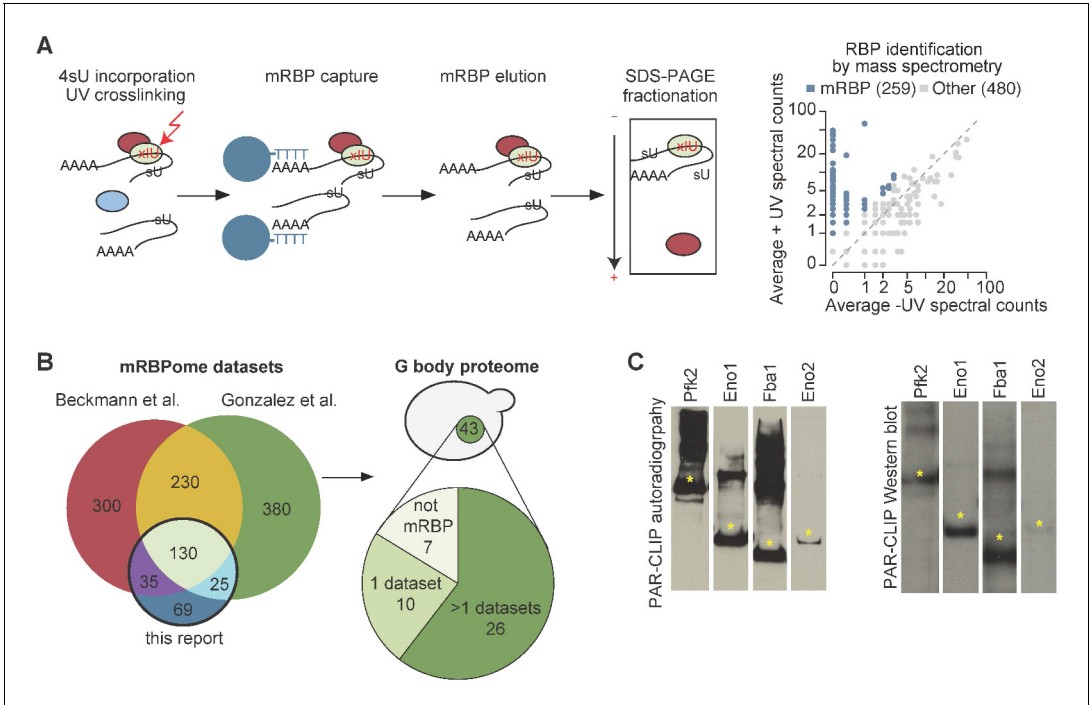

**Figure 1.** G bodies are enriched for RNA-binding proteins. (**A**) Photoactivatable-ribonucleoside-enhanced cross-linking with oligo(dT) affinity purification and tandem mass spectrometry (PAR-CL-MS) pipeline. Briefly, 4-thiouridine (4SU) is incorporated by supplementation in media and crosslinked to protein with 365 nm UV light. mRBPs are captured with oligo-d(T) beads and eluted. RNPs are fractionated by SDS-PAGE and resulting proteins are analyzed by proteomic mass spectrometry. Average mass spectrometry spectral peak counts from biological replicate PAR-CL-MS versus non-UV-treated control. Proteins enriched in PAR-CL-MS (blue dots) had either (1)>2 spectral counts in PAR-CL and 0 spectral counts in -UV control or (2) had FDR < 10% as calculated by QSPEC (*Choi et al., 2008*). (**B**) Overlap of identified mRBPs with datasets generated with similar methodology. mRBP datasets are from *Matia-González et al. (2015)* and *Beckmann et al., 2015*. Distribution of G body proteome (union of colocalization validated G-body proteins from *Jin et al., 2017* and *Miura et al., 2013*. (**C**) Autoradiography and western blot of PAR-CLIP of TAP-tagged proteins (Pfk2, Eno1, Eno2, and Fba1). Stars indicate the same band in autoradiography and western blot confirming RNA binding by each indicated protein.

The online version of this article includes the following source data and figure supplement(s) for figure 1:

**Figure supplement 1.** Identification of glycolysis enzyme binding sites.

**Figure supplement 1—source data 1.** Source data for *Figure 1—figure supplement 1E*.

(*Beckmann et al., 2015*; *Matia-González et al., 2015*). Of the RNA binding glycolysis enzymes, 10 localize to G bodies in hypoxia (*Figure 1—figure supplement 1A*), (*Jin et al., 2017*).

## Glycolysis enzymes bind similar transcripts

To validate RNA binding by glycolysis enzymes, we end-labeled potential RNAs crosslinked to TAP-tagged Pfk2, Eno1, Eno2, and Fba1 (*Figure 1C*, left panels) with $\gamma-32$-P ATP. The autoradiographs displayed the same migration pattern in SDS-PAGE as the immunoblots of the corresponding TAP-tagged glycolysis enzymes (*Figure 1C*, right panels), suggesting that the glycolysis enzymes Pfk2, Eno1, Eno2, and Fba1 bind RNA. We then identified the mRNA substrates of Pfk2, Eno1, and Fba1 by performing photoactivatable, ribonucleoside-enhanced cross-linking and immunoprecipitation (PAR-CLIP) followed by deep sequencing (PAR-CLIP-seq; *Hafner et al., 2010*) in log phase yeast cells grown under normoxic conditions. To determine the binding site sequences, we empirically determined 'high-confidence' RPM (reads per million mapped reads) thresholds (Pfk2: 5 RPM, Fba1: 0.5 RPM, Eno1: 0.5 RPM) for each library (*Figure 1—figure supplement 1B*). We identified 1540 total mRNAs that bind at least one of these three glycolysis enzymes. Specifically, there were 439 direct mRNA substrates of Pfk2 with 559 discrete binding sites, 1,001 mRNA substrates with 1432 binding sites for Eno1, and 721 mRNA substrates with 1014 binding sites for Fba1 (*Figure 1—figure supplement 1B*, right panels). The results of Eno1 PAR-CLIP-seq were in agreement with a recent analysis of Eno1 in normoxic conditions by CRAC (a method that UV crosslinks and affinity purifies

protein-RNA complexes under denaturing conditions *Shchepachev et al., 2019*), providing additional validation of our identified binding sites. We identified 69 out of the top 100 bound mRNAs in the CRAC dataset. mRNAs bound by each glycolysis enzyme displayed substantial overlap, with 490 mRNAs binding at least two of the three glycolysis enzymes and 131 mRNAs binding all three (*Figure 2A*). Intriguingly, bound mRNAs of all three enzymes were enriched for functional annotations related to glycolysis as well as other metabolic pathways (*Figure 2B*). For example, Pfk2 targets included mRNAs encoding 13 of the 22 known glycolytic enzymes (*Figure 1—figure supplement 1C*). Fba1 and Eno1 also bound to mRNAs encoding glycolysis enzymes, and together, Pfk2, Fba1, and Eno1 bound to 16 of the 22 glycolytic enzyme-encoding mRNAs in yeast (*Figure 1—figure supplement 1C*).

With the exception of Fba1-bound mRNAs, which are primarily in non-coding RNAs and coding sequences, most of the glycolytic enzyme binding sites on substrate mRNAs were within the 3' untranslated regions (3' UTRs) and coding sequences, followed by non-coding RNAs and 5' UTRs (*Figure 2C*). We used the high-confidence binding sites for each glycolysis enzyme to identify enriched motifs. Pfk2 binding sites contained an AU-rich element, similar to elements that regulate mRNA stability in yeast (*Vasudevan and Peltz, 2001*), whereas Eno1 and Fba1 binding sites contained pyrimidine-rich motifs (*Figure 2D*). Furthermore, the binding sites for each enzyme displayed overlap between enzymes. Although binding footprints were short (22 nt on average), between 10–17% of binding sites were bound by at least two of the three glycolysis enzymes (Fba1, Eno1, or Pfk2), and 13–19% of the sites for one enzyme partially overlapped with the binding sites of at least one other glycolysis enzyme with the remainder of binding sites being uniquely bound by either Pfk2, Eno1 or Fba1 (*Figure 2E*). These partially overlapping sites and identical shared sites had greater average sequencing depth than unique sites bound by only one glycolysis enzyme (*Figure 2F*). Unique binding sites, however, had a greater log enrichment of binding site RPM to gene RPKM (reads per kilobase of transcript per million mapped reads) than overlapping sites and identical sites in multiple datasets, suggesting that these sites were more tightly bound (*Figure 2F*). Thus, the greater binding frequency on overlapping sites was largely driven by the amount of target mRNA. Nevertheless, the overlap in bound transcripts and binding sites suggests that common RNA binding could contribute to the coalescence of glycolysis enzymes into G bodies.

## G bodies copurify with and contain RNAs

We previously developed a method using differential centrifugation and affinity capture to isolate G bodies and identify their resident proteins by mass spectrometry, followed by validation using colocalization to G bodies (*Jin et al., 2017*). Of the 43 identified G body components, 36 are mRBPs (*Jin et al., 2017*; *Miura et al., 2013*; *Figure 1B*, *Supplementary file 1*). Glycolysis enzymes are diffusely localized in the cytosol of yeast cells and bind RNA under normoxic growth conditions (*Figure 2*). The observation that many of the proteins targeted to G bodies upon shift to hypoxic conditions bind RNA under normoxic growth, including glycolysis enzymes, raised the possibility that G bodies themselves contain RNA. To test this hypothesis, we adapted our previous G body purification protocol to perform RNA extraction, cDNA synthesis, and quantitative PCR to evaluate the presence of co-purifying RNA (*Figure 2—figure supplement 1A*). In order to reduce background, we generated a yeast strain with an integrated Pfk2-GFP-1xFlag transgene. This yeast strain was shifted to hypoxic conditions for 18 hr to induce G body formation. G bodies were then immunopurified from lysates with a monoclonal mouse anti-Flag antibody and eluted with Flag peptide before proteinase K digestion and RNA extraction (*Figure 2—figure supplement 1A,B*). As a control for nonspecific RNA binding, we performed the same protocol with wild type BY4742 cells and extracted RNA from the flow through for each sample (*Figure 2—figure supplement 1A,B*). By comparing the relative amount of RNA from the flow through and the eluate, we determined the percent of RNA in the eluate compared to the input of the immunoprecipitation. We tested a range of qPCR probes for RNAs with or without binding sites identified in PAR-CLIP-seq of Eno1, Pfk2 and Fba1 (*Supplementary file 2*). While Pfk2-GFP-Flag pulldowns recovered between 2.4% and 7% of the flow-through RNA, we recovered at most 1.5% of flow-through RNA in control experiments (*Figure 2—figure supplement 1C*), indicating a 3.5 to 11.5-fold enrichment of these RNAs in G bodies versus the BY4742 control. These trends were not different for RNAs with binding sites for glycolysis enzymes, suggesting that RNA binding by other RBPs in G bodies may contribute to RNA accumulation in G bodies. However, the low recovery rate relative to flow-through RNAs suggests that only a

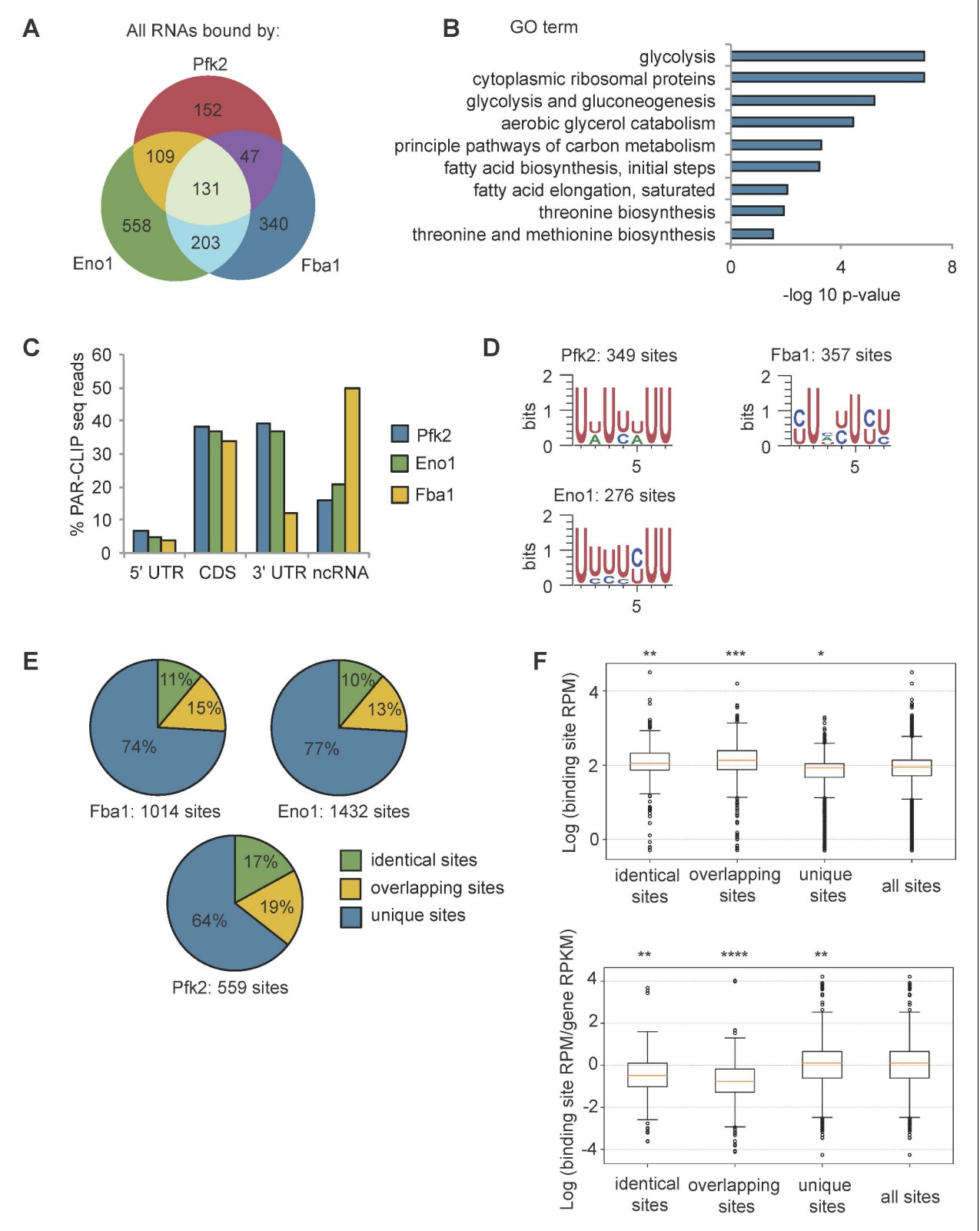

**Figure 2.** Glycolysis enzymes bind similar RNAs. (**A**) Overlap of mRNAs bound by Pfk2, Eno1, and Fba1 as identified by PAR-CLIP-seq in normoxic conditions. (**B**) Gene ontology (GO) terms enriched among transcripts containing high-confidence Pfk2, Eno1 and Fba1 sites include glycolysis. Fisher's exact test *p*-values are plotted. (**C**) Percent of total Pfk2, Eno1, and Fba1 PAR-CLIP-seq reads per million mapped reads (RPM), aligning with the indicated genic regions. (**D**) Identified sequence motifs among Pfk2, Eno1, and Fba1 binding sites. (**E**) Percent of binding sites for each PAR-CLIP-seq

*Figure 2 continued on next page*

*Figure 2 continued*

dataset that are bound by more than one glycolysis enzyme (identical sites), overlap a binding site of another glycolysis enzyme (overlapping sites), or are bound by only one glycolysis enzymes (unique sites) (F) (Top) Binding sites present in multiple datasets or overlapping other binding sites tend to have greater read depth. Boxplot of $\log_{10}$ (Binding Site RPM) for each class of binding site. *p*-values represent unpaired student's T Tests. (Bottom) Glycolysis enzymes tend to bind tighter to unique sites. Boxplot of $\log_{10}$ ratio of binding site RPM to gene RPKM from mRNA seq. *p*-values represent unpaired student's T tests. *$p<10^{-5}$. **$p<10^{-7}$. ***$p<10^{-9}$. ****$p<10^{-20}$.

The online version of this article includes the following source data and figure supplement(s) for figure 2:

**Figure supplement 1.** G bodies co-purify with RNA.

**Figure supplement 1—source data 1.** Source data for *Figure 2—figure supplement 1C*.

small fraction of each mRNA accumulates in G bodies, unlike stress granules, where up to 95% of particular mRNAs localize to stress granules (*Khong et al., 2017*).

Given the strong enrichment relative negative controls, we determined the global distribution of mRNAs by sequencing RNA from immunopurified G bodies (RIP-seq), as well as the RNA in the flow-through and total RNA. Gene RPMs within individual replicates were highly correlated (*Figure 2—figure supplement 1D*), thus we averaged each replicate. To determine G body enriched transcripts, we calculated the fold enrichment over flow-through RNA and total RNA. Applying a threshold of two-fold enrichment with a *p*-value of less than 0.05, we identified 297 transcripts enriched in purified G bodies over the flow-through RNA (*Figure 3A*, *Supplementary file 4*), most of which were also enriched over total RNA (*Figure 3B*, *Supplementary file 4*). Applying the same criteria for both enrichment over flow-through and enrichment over total RNA left us with 101 high-confidence G body RNAs. Strikingly, the number of RNAs bound by glycolysis enzymes in these 101 G body RNAs was nearly 2-fold greater than expected by chance (*Figure 3C,D*), indicating a correlation between the RNAs associated with G bodies under hypoxia and the RNAs that bind to diffusely localized glycolysis enzymes under normoxic conditions, suggesting that RNA binding to glycolysis enzymes is important for localization of mRNA to G bodies.

To validate the localization of RNA to G bodies and the validity of our G body RIP-seq, we performed smFISH of five transcripts with varying enrichment in G bodies and analyzed localization relative to several G body reporters (Eno2-GFP, Fba1-GFP and Pfk2-GFP). In each example, we detected cases of overlap and peripheral localization of RNA to G bodies (*Figure 3E*, *Figure 3—figure supplement 1A,B*). The number of transcripts per cell was similar across replicates and strongly correlated with total RNA abundance, validating our smFISH analysis (*Figure 3—figure supplement 1C,D*). To determine the degree to which G bodies colocalized with RNA, we measured the distance from G body centers to each mRNA detected in the same focal plane as the G body (*Figure 3—figure supplement 1E*). The percent of mRNAs within a single pixel (0.13 μm) of a G body center correlated with enrichment in G body IPs, suggesting that G body RIP-seq reflects association of mRNA with G bodies (*Figure 3E*). However, this represented only a small fraction of the total pool of mRNA, with maximally 2% of a particular mRNA being localized to G bodies. Since the *OLE1* mRNA was not enriched in G body RIP-seq, we used it as a negative control when analyzing distance from G body centers. Normalizing the cumulative distribution of distances from G body centers for each mRNA to the *OLE1* distribution revealed that G body mRNAs were most enriched near G body centers, becoming more randomly distributed as the distance from G bodies increased (*Figure 3G*). We also noticed that there was often a cluster of mRNAs around the periphery of G bodies (*Figure 3E*, *Figure 3—figure supplement 1A*). This peripheral localization is reflected in smaller peaks in the normalized cumulative distribution for each mRNA between 0.5 and 1 μm from the G body center, indicating that this localization pattern is specific to the enriched mRNAs. Taken together, these data suggest G bodies induced by hypoxia contain mRNAs that are initially bound by glycolysis enzymes in normoxia, but only concentrate a small fraction of any given mRNA to G bodies.

## Targeting RNase to nascent G body sites inhibits formation in vivo

Given that G bodies are enriched for RNA binding proteins and G body resident glycolysis enzymes share many common mRNA substrates, we next tested whether G body formation requires RNA in vivo. We targeted an RNAse to nascent G bodies by fusing Pfk2 to *E. coli* MqsR, an RNase that preferentially cleaves RNA at GCU sequences, followed by a C-terminal Flag tag (*Kasari et al., 2010*;

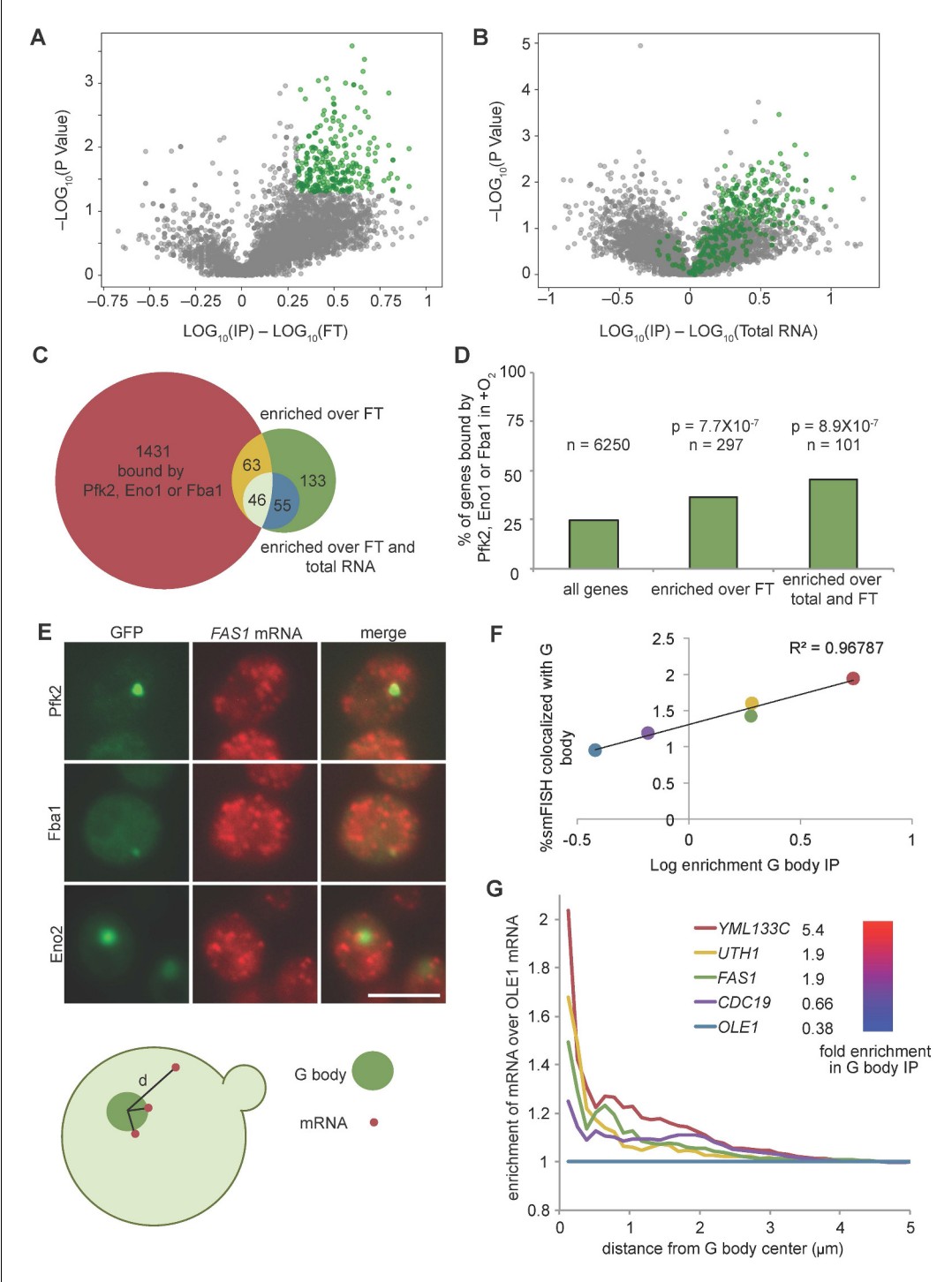

**Figure 3.** G body associated RNAs overlap with RNAs bound by glycolysis enzymes in normoxia. (**A**) Volcano plot of RNAs enriched in G body RIPs over flow through RNAs. (**B**) Volcano plot of RNAs comparing G body RIPs and total RNA. (**A–B**) RNAs highlighted in green represent RNAs > 2 fold enriched over flow through with p<0.05 (T Test), n = 297. (**C**) Overlap of enriched RNAs with RNAs identified by PAR-CLIP binding to Eno1, Fba1 or Pfk2. Enriched RNAs are >2 fold enriched in RPM in G body RIPs with p<0.05 (T Test) comparing to indicated datasets. (**D**) Bar graph of % of genes mRNAs with normoxic binding sites for Eno1, Fba1 or Pfk2. *P*-value is Chi squared test. (**E**) smFISH representative images of Pfk2-GFP, Fba1-GFP and Eno2-GFP colocalization with *FAS1* mRNA. Scale bar is 5 μm. (Bottom) Schematic of distance measurement to determine colocalization. Shortest distance from G body center to mRNA was measured. (**F**) Percent of mRNAs in a single Z slice within one pixel of the G body center determined by Eno2-GFP, Fba1-GFP or Pfk2-GFP plotted against log₁₀ enrichment in G body RIP-seq over total RNA. Average and standard deviation of two

*Figure 3 continued on next page*

*Figure 3 continued*

biological replicates plotted. (**G**) Fold change of cumulative distribution of distances from G body center for each mRNA tested over *OLE1* mRNA. n = 3773 *CDC19*, 3781 *FAS1*, 3532 *UTH1*, 3036 *YML133C* and 5159 *OLE1* mRNAs.

The online version of this article includes the following source data and figure supplement(s) for figure 3:

**Source data 1.** Source data for *Figure 3C,D*.
**Source data 2.** Source data for *Figure 3F*.
**Source data 3.** Source data for *Figure 3G*.
**Figure supplement 1.** Validation of smFISH colocalization with G body markers.
**Figure supplement 1—source data 1.** Source data for *Figure 3—figure supplement 1C*.
**Figure supplement 1—source data 2.** Source data for *Figure 3—figure supplement 1D*.
**Figure supplement 1—source data 3.** Source data for *Figure 3—figure supplement 1E*.

*Yamaguchi et al., 2009*). We placed Pfk2-MqsR-Flag under the control of the copper sulfate (CuSO$_4$) inducible *CUP1* promoter on a centromeric plasmid, and introduced this plasmid, or a control vector, into cells expressing the G body reporter Pfk2-GFP (*Figure 4A*). We detected weak expression of Pfk2-MqsR-Flag even in the absence of CuSO$_4$, consistent with weak activation of the *CUP1* promoter in hypoxic conditions (*Becerra et al., 2002*). Cells treated with CuSO$_4$ showed a dose-dependent increase in Pfk2-MqsR-Flag levels (*Figure 4—figure supplement 1A*, top panels). At low concentrations of CuSO$_4$ (5, 10 μM) in hypoxia, the Pfk2-MqsR-Flag fusion protein was targeted to G bodies and colocalized with Pfk2-GFP (*Figure 4—figure supplement 1B,C*). Using this same approach, another RNase, RNase A, could also be induced and targeted to G bodies as a Pfk2-RNase A fusion protein (*Figure 4—figure supplement 1D,E*).

Targeting Pfk2-MqsR-Flag to G bodies caused a robust reduction in the fraction of hypoxic cells with G bodies in a dose-dependent manner (*Figure 4B*). In contrast, G body formation in hypoxic cells carrying a control vector was unaffected by CuSO$_4$ treatment at all concentrations tested (*Figure 4B*). Even without induction, hypoxic cells carrying the Pfk2-MqsR-Flag plasmid showed a 20% decrease in G body formation, consistent with the weak expression of Pfk2-MqsR-Flag under these conditions. CuSO$_4$-induced expression of Pfk2-RNase A similarly inhibited G body formation (*Figure 4—figure supplement 2A*).

The MqsR RNase is inhibited by its antitoxin, MqsA (*Kasari et al., 2010*; *Yamaguchi et al., 2009*). To verify that MqsR nuclease activity was required to inhibit G body formation, we fused the antitoxin MqsA to the Pfk2-MqsR-Flag construct, generating a Pfk2-MqsR-Flag-MqsA fusion protein. At both 0 and 50 μM CuSO$_4$, G body formation in hypoxic cells was now unaffected by induction of Pfk2-MqsR-Flag-MqsA (*Figure 4C*), indicating that MqsA effectively inhibited the G body-targeted MqsR RNase from disrupting G body formation. G bodies appeared larger and brighter in Pfk2-MqsR-Flag-MqsA-expressing cells than in cells expressing a vector control, possibly due to overexpression of Pfk2 (*Figure 4C*). Similarly, a Pfk2-RNase A$^{H12A}$ mutant with decreased RNase activity was less effective at reducing G body formation than Pfk2-RNase A$^{wild-type}$, especially at 100 μM CuSO$_4$ (*Figure 4—figure supplement 2B*). Pfk2-RNase A$^{H12A}$ possesses residual RNase activity (*Thompson and Raines, 1994*), which likely contributes to the modest loss of G body formation. Thus, inhibition of G body formation by RNase A and MqsR is due to their ribonuclease activity.

To further test whether specific targeting of MqsR-Flag to G bodies by fusion to Pfk2 was required for inhibiting G body formation, we engineered cells expressing MqsR-Flag alone, which was inducible by addition of CuSO$_4$ (*Figure 4—figure supplement 1A*, bottom panels). Unlike Pfk2-MqsR-Flag expression, MqsR-Flag expression had no effect on G body formation in hypoxic cells (*Figure 4D*). In addition, we tested the ability of Pfk2-RNase A to inhibit G body formation of several other G body markers, including Eno2, Cdc19, and Fba1. In each case, CuSO$_4$ induction of hypoxic cells expressing Pfk2-RNase A, but not the vector control, led to a decrease in G body formation in a CuSO$_4$ concentration-dependent manner (*Figure 4—figure supplement 3A*). Expression of RNase alone or loss of RNase activity prevents inhibition of G body formation. Taken together, directing a functional RNase to sites of G body formation leads to loss of G bodies, indicating that RNA is required for G body formation in vivo (*Figure 4E*).

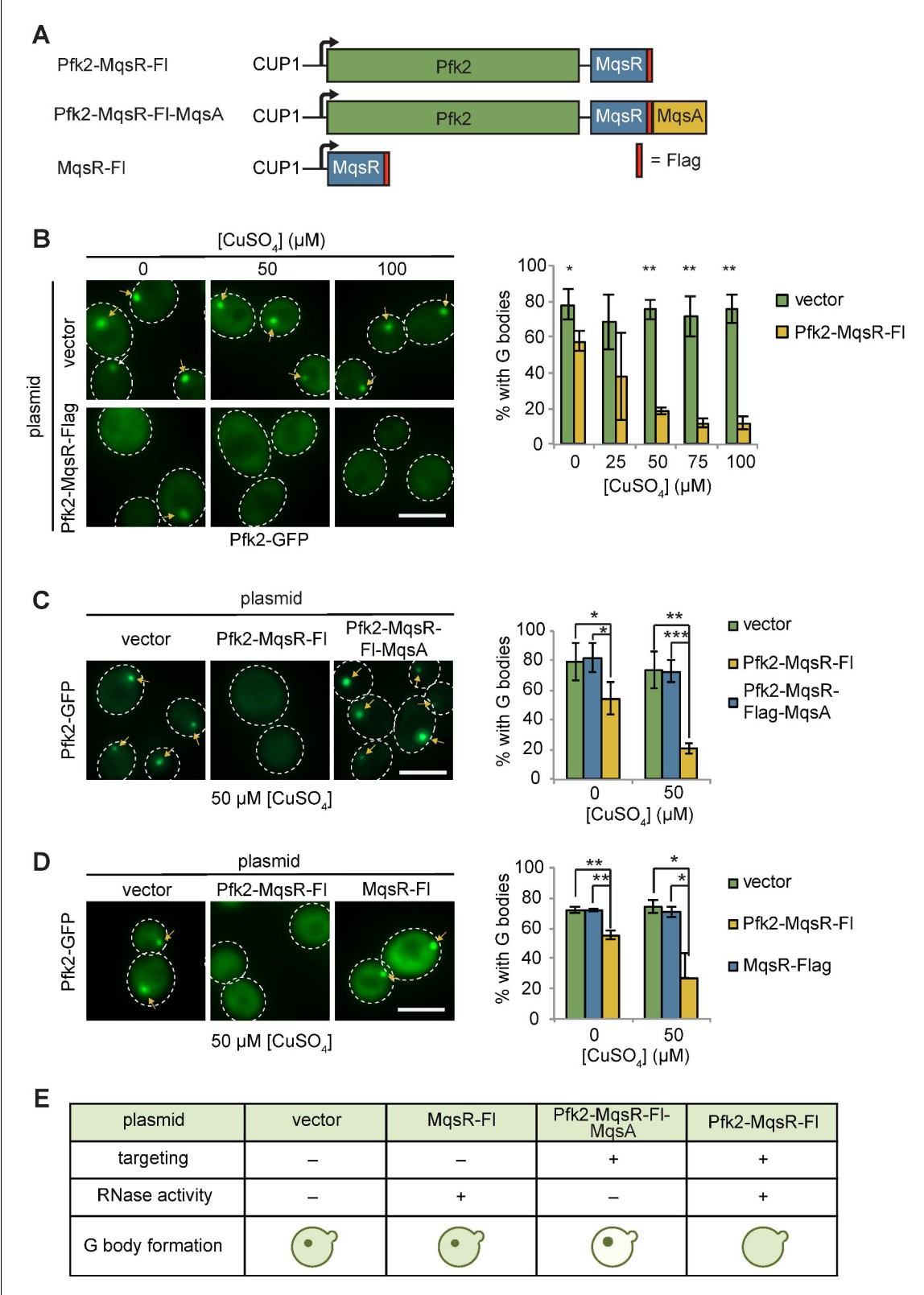

**Figure 4.** Tethering Pfk2 to an RNase prevents G-body formation. (**A**) Schematic of Pfk2-MqsR constructs. All constructs were expressed from a centromeric plasmid under control of the $CuSO_4$-inducible *CUP1* promoter. (**B**) (Left) Representative images of hypoxic Pfk2-GFP localization with increasing concentrations of $CuSO_4$ for cells expressing a vector control or cells expressing Pfk2-MqsR-Flag. (Right) Quantification of G body formation in cells with varying $CuSO_4$ concentrations. (**C**) (Left) Representative images of hypoxic Pfk2-GFP localization comparing cells with a vector control, Pfk2-

*Figure 4 continued on next page*

*Figure 4 continued*

MqsR-Fl, or Pfk2-MqsR-Fl-MqsA with 50 µM CuSO₄. (Right) Quantification of G body formation for cells with each plasmid with 0 and 50 µM CuSO₄. (D) (Left) Representative images of hypoxic Pfk2-GFP localization at 50 µM CuSO₄. (Right) Quantification of G body formation for cells with each plasmid with 0 and 50 µM CuSO₄. (E) Schematic summary of results of G body formation with either a vector plasmid or plasmids expressing either MqsR-Flag, Pfk2-MqsR-Flag, or Pfk2-MqsR-Flag-MqsA. All scale bars are 5 µM. For each graph, data represent mean and standard deviation of three to four individual experiments (n > 100 cells per replicate per condition). Arrows indicate G bodies. Statistics were analyzed by unpaired student's T tests with a Bonferroni correction for multiple testing. *p<0.05. **p<0.01. ***p<0.001.

The online version of this article includes the following source data and figure supplement(s) for figure 4:

**Source data 1.** Source data for *Figure 4B*.
**Source data 2.** Source data for *Figure 4C*.
**Source data 3.** Source data for *Figure 4D*.
**Figure supplement 1.** Validation of RNase tagged Pfk2 variants.
**Figure supplement 1—source data 1.** Source data for *Figure 4—figure supplement 1C*.
**Figure supplement 2.** Pfk2-RNase A inhibits G-body formation.
**Figure supplement 2—source data 1.** Source data for *Figure 4—figure supplement 2A*.
**Figure supplement 2—source data 2.** Source data for *Figure 4—figure supplement 2B*.
**Figure supplement 3.** Pfk2-RNase A inhibits punctate formation of multiple G-body markers.
**Figure supplement 3—source data 1.** Source data for *Figure 4—figure supplement 3A*.
**Figure supplement 3—source data 2.** Source data for *Figure 4—figure supplement 3B*.
**Figure supplement 3—source data 3.** Source data for *Figure 4—figure supplement 3C*.

## Targeting RNase to pre-formed G bodies results in multiple puncta formation

To test whether RNA was required for the stability of G bodies that had already formed, cells were first grown in hypoxic conditions for 20 hr and then shifted to normoxic conditions along with Pfk2-MqsR-Flag induction by addition of CuSO₄. Immediately following shift from hypoxic to normoxic conditions and concomitant CuSO₄ addition (i.e., '0 hr post-hypoxia' in *Figure 5A*; *Figure 5B*), Pfk2-MqsR-Flag carrying cells showed a modest reduction in G body formation compared to controls (*Figure 5A,B*), likely due to leaky expression of Pfk2-MqsR-Flag in hypoxia. However, we reasoned that we could still test the effects of inducing Pfk2-MqsR, as most cells (61%) had G bodies. After prolonged induction of Pfk2-MqsR-Flag in normoxic conditions ('20 hr post-hypoxia' in *Figure 5D*; *Figure 5E*), we observed strong expression of Pfk2-MqsR-Flag at 100 µM CuSO₄ and low-level expression even in cells without addition of CuSO₄ (*Figure 5C*). Under these conditions, cells harboring a vector control either had a single G body per cell that persisted or diffuse localization of Pfk2-GFP, regardless of added CuSO₄. In contrast, a substantial proportion of cells (41–50%) with Pfk2-MqsR-Flag showed multiple puncta per cell (*Figure 5D,E*). These data suggest that pre-formed G bodies fracture into multiple structures in the presence of Pfk2-MqsR-Flag.

Taken together, these data indicate that 1) Nascent G body formation was inhibited by RNases in a concentration-dependent manner (*Figure 4A*); 2) Inhibition of de novo G body formation required both RNase activity and targeting to the site of G body formation (*Figure 4C–E*); 3) Once formed, targeting an RNase to G bodies led to cells with multiple foci, suggesting fragmentation but not dissolution of G bodies (*Figure 5D–F*); and 4) Once formed in hypoxia, G bodies could persist for tens of hours in normoxia (*Figure 5D–F*). Despite the fact that untargeted MqsR does not decrease G body formation (*Figure 4D*) and targeted nucleases localize to G bodies when weakly induced (*Figure 4—figure supplement 1B,E*), it is formally possible that loss of expression of specific mRNAs bound by Pfk2 leads to reduced G body formation by reducing levels of G body components or by some unknown mechanism. However, loss of G body components is unlikely to explain the fracturing of extant G bodies following induction after shifting from hypoxia to normoxia.

## G body recruitment requires multivalent interactions

IDRs have been shown to modify phase separation behavior in vitro and in vivo (*Kato et al., 2012*; *Mitrea and Kriwacki, 2016*; *Molliex et al., 2015*; *Protter et al., 2018*). Pfk2 has such an IDR in its N-terminal domain (*Figure 6A*, blue box). Our previous study demonstrated that deletion of an N-terminal IDR spanning amino acids 140–165 of Pfk2-GFP (Pfk2$^{\Delta140\text{-}165}$-GFP) increased the number of cells forming multiple foci and decreased overall G body formation (*Jin et al., 2017*). The Pfk2

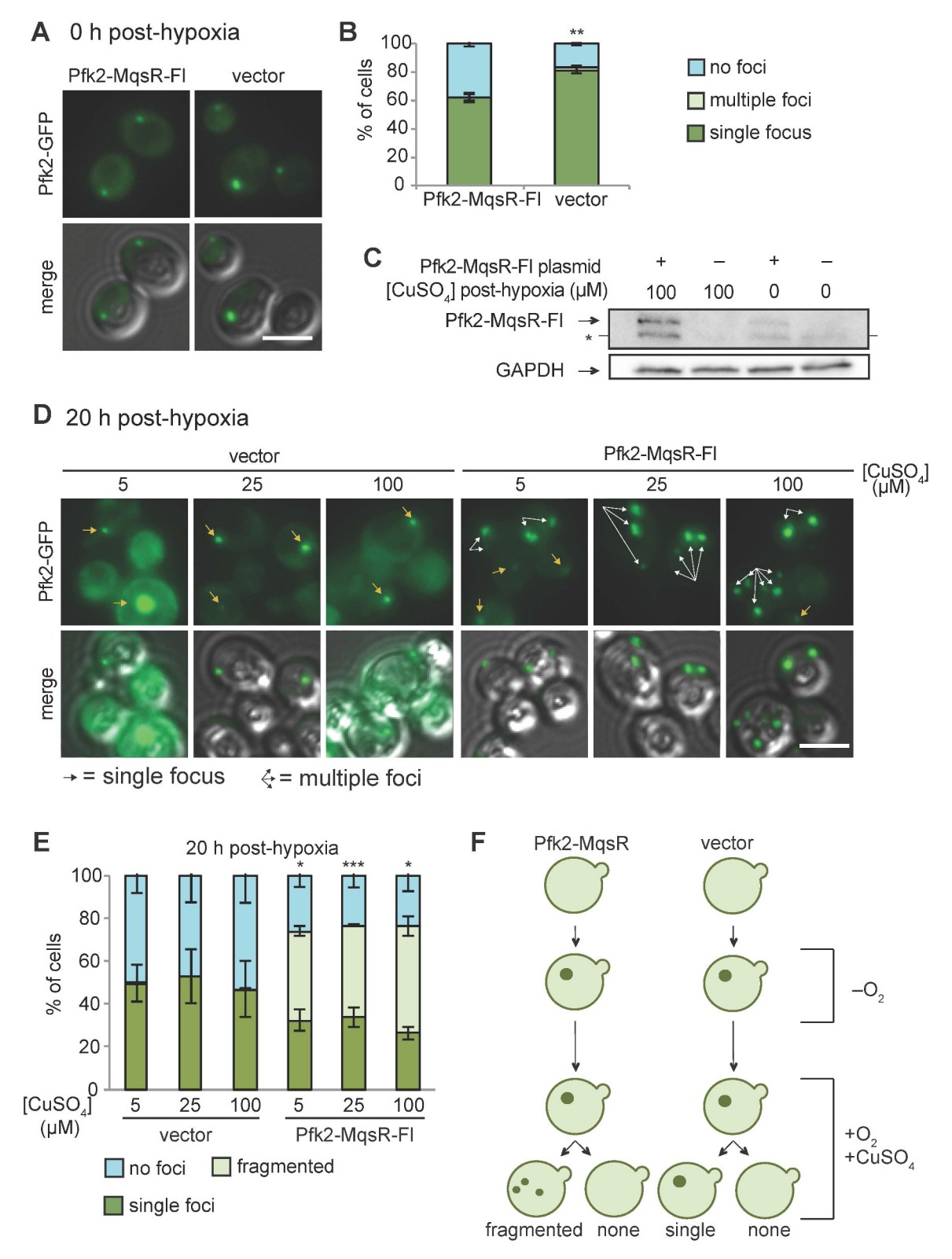

**Figure 5.** Pfk2-MqsR-Fl induction fractures existing G bodies. (**A**) Cells with a plasmid inducibly expressing Pfk2-MqsR-Flag form G bodies in the absence of induction by CuSO₄. Representative images of hypoxic Pfk2-GFP localization in cells expressing a vector control or Pfk2-MqsR-Fl with 0 μM CuSO₄. (**B**) Quantification of G body formation of cells in (**A**). (**C**) Western blot showing induction of Pfk2-MqsR-Fl where CuSO₄ is added after 20 hr in hypoxia and cells are subsequently cultured for 20 hr in normoxia. Pfk2-MqsR-Fl is probed with a monoclonal anti-Flag antibody. GAPDH serves as a

*Figure 5 continued on next page*

*Figure 5 continued*

loading control. * indicates a nonspecific band. (D) Representative images of Pfk2-GFP localization in cells expressing a vector control and cells expressing Pfk2-MqsR-Fl after 20 hr hypoxia followed by induction with varying concentrations of CuSO$_4$ in normoxia for 20 hr. Cells with Pfk2-MqsR-Fl frequently have multiple large foci. (E) Quantification of cells with a single focus, multiple foci, or no foci for cells in (D). (F) Cartoon showing Pfk2-GFP localization before hypoxia, immediately after hypoxia, and after induction with CuSO$_4$. All graphs show mean and standard deviation of three independent experiments (n > 100 cells per condition per replicate). Arrows indicate G bodies. Statistics calculated with unpaired student's T tests. All scale bars 5 µm. *p<0.05. **p<0.01. ***p<0.001.

The online version of this article includes the following source data for figure 5:

**Source data 1.** Source data for *Figure 5B*.
**Source data 2.** Source data for *Figure 5E*.

N-terminal domain is conserved in *S. cerevisiae* (28.7% identity) and other yeast species. The crystal structure of *S. cerevisiae* heterooctameric phosphofructokinase (with four copies each of Pfk1 and Pfk2) lacks this region for both Pfk1 and Pfk2, due to proteolytic cleavage in sample preparation (*Banaszak et al., 2011*). However, the crystal structure of phosphofructokinase from a related yeast species, *Komegataella pastoris* (*Sträter et al., 2011*), shows that the Pfk2 N-terminal domain resembles a glyoxylase domain and interacts with Pfk1. Regions outside of the Pfk2 N-terminus also interact with Pfk1 in both crystal structures to form a heterooctamer with four Pfk2 and four Pfk1 molecules. These multiple interaction domains provide the potential for multivalent Pfk2 interactions.

To test whether N-terminal interactions are important for Pfk2 recruitment to G bodies, we deleted its N-terminal domain (Δ1–202) (*Figure 6B*). This region encompasses the most disordered residues including the previous IDR deletion (Δ140–165) (*Figure 6B*). Compared with full-length Pfk2-GFP, cells expressing Pfk2$^{\Delta1-202}$-GFP had drastically reduced puncta (*Figure 6C*), suggesting that the N-terminal domain is required for either G body formation or Pfk2 localization to G bodies. As a control, we retested the Pfk2$^{\Delta140-165}$-GFP mutant and observed decreased G body localization with an increased instance of cells with multiple foci. Notably, the localization defect was much stronger in Pfk2$^{\Delta1-202}$-GFP cells, suggesting that the structured portions of the N-terminal domain act in concert with the disordered residues to promote localization to G bodies.

To test whether the N-terminal region is sufficient for G body localization, we fused the N-terminal region alone to GFP (Pfk2$^{1-202}$-GFP, *Figure 6B*). When *PFK2$^{1-202}$-gfp* was integrated and replaced full-length endogenous *PFK2*, Pfk2$^{1-202}$-GFP did not form puncta (*Figure 6D*). However, when Pfk2$^{1-202}$-GFP was expressed from a plasmid in wild-type cells expressing endogenous full-length Pfk2, we detected robust puncta formation (*Figure 6D*). These data suggest that the N-terminal region of Pfk2 is not sufficient for G body localization, but requires interaction with full-length Pfk2 for recruitment to G bodies. To test the importance of the ordered interactions by the remainder of the protein, we probed Pfk1 recruitment to G bodies in wild-type and *pfk2Δ* cells. The frequency of hypoxic cells with Pfk1-GFP puncta was reduced in *pfk2Δ* cells compared to wild-type, suggesting that Pfk1 is recruited to G bodies in concert with Pfk2 (*Figure 6E*).

## G bodies behave as gels

To determine if G bodies behave as liquid-like structures or more solid gels, we treated hypoxic cells with 5% 1,6-hexanediol. The alcohol 1,6-hexanediol specifically dissolves liquid-like structures purportedly due to its ability to disrupt weak hydrophobic interactions, and therefore has been used to discern liquid-like from solid-like granules (*Kroschwald et al., 2017*; *Kroschwald et al., 2015*). We noticed no significant loss of G bodies after 1 hr treatment, whereas untreated cells showed a modest loss of G bodies (*Figure 7—figure supplement 1A*). The loss of G bodies in the control condition was likely due to continued cell division, which is inhibited by 1,6-hexanediol (*Kroschwald et al., 2017*). However, G bodies in 1,6-hexanediol treated cells appeared smaller than those in untreated cells. Therefore, we quantified the size of G bodies by fitting 2-dimensional Gaussian distributions to maximum intensity projections of G bodies in untreated and 1,6-hexanediol treated cells and computed the parameters of the fit. In 1,6-hexanediol treated cells, the average $\sigma_X$ (0.72 +/– 0.055 µm) and $\sigma_Y$ (0.74 +/– 0.060 µm) values were lower than untreated cells (1.1 +/– 0.070 µm and 1.1 +/– 0.067 µm, respectively), indicating a smaller granule radius (*Figure 7—figure*

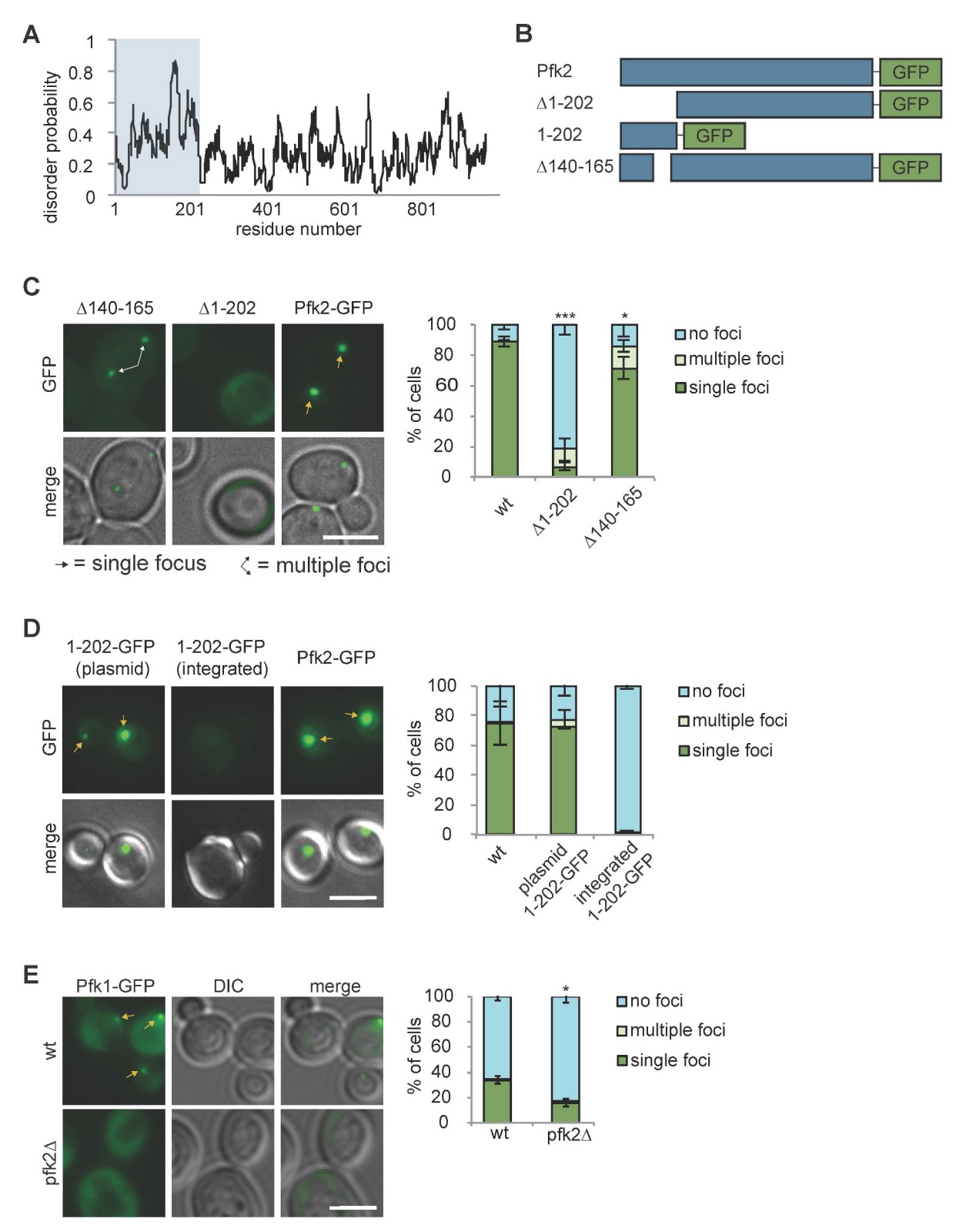

**Figure 6.** Protein recruitment to G bodies relies on multivalent interactions. (**A**) IUPRED prediction of disorder in Pfk2. The first 202 amino acid residues (i.e., the Pfk2 N-terminus) are largely unstructured. (**B**) Schematic of Pfk2-GFP variants tested. (**C**) (Left) The Pfk2 N-terminus is required for G body localization. Representative images of Pfk2-GFP variant localization in hypoxia. (Right) Quantification of G body localization. (**D**) (Left) The Pfk2 N-terminus is not sufficient for G body localization. Representative images of Pfk2-GFP variant localization in hypoxia. When integrated (in the absence

*Figure 6 continued on next page*

*Figure 6 continued*

of full-length Pfk2), GFP fused to the Pfk2 N-terminus does not localize to G bodies. When expressed on a plasmid in a strain expressing full-length Pfk2, GFP fused to the Pfk2 N-terminus is able to localize to G bodies. (Right) Quantification of G body localization. (E) (Left) Pfk1-GFP recruitment to G bodies depends on Pfk2. Representative images of hypoxic Pfk1-GFP localization in wild-type and *pfk2Δ* cells. (Right) Quantification of Pfk1-GFP localization in wild-type and *pfk2Δ* cells in hypoxia. All scale bars are 5 μm. Arrows represent either G bodies or cells with multiple G bodies. For (C) and (E): Data represent mean and standard deviation of three replicates (n > 100 cells per condition per replicate). For (D): Data represent mean and standard deviation of two biological replicates (n > 100 cells per condition per replicate). Statistics are unpaired student's T tests comparing G body formation. *p<0.05. **p<0.01. ***p<0.001.

The online version of this article includes the following source data for figure 6:

**Source data 1.** Source data for *Figure 6C*.
**Source data 2.** Source data for *Figure 6D*.
**Source data 3.** Source data for *Figure 6E*.

---

*supplement 1A*). Additionally, the amplitude was lower in the 1,6-hexanediol treated cells (12,054 +/– 857 A.U. compared to 18,519 +/– 1308 A.U.), indicating reduced total fluorescence in the granules. Thus, while 1,6-hexanediol does not fully dissolve G bodies, it can reduce their size. As a control, we verified that 1,6-hexanediol treatment of glucose-starved cells disrupted P bodies labeled with Edc3-GFP, but caused only a small and not significant loss of stress granules labeled with Pab1-GFP (*Kroschwald et al., 2015*; *Figure 7—figure supplement 1B*). To test whether an early liquid-liquid phase separation (LLPS) state precedes G body formation, we treated cells with 1,6-hexanediol during nascent granule formation. Because high concentrations (5%) of 1,6-hexanediol severely inhibited cell growth, we tested G body formation with 2% 1,6-hexanediol. As a control, we verified that addition of 1,6-hexanediol to the starvation media inhibited the formation of P bodies but not stress granules (*Figure 7—figure supplement 1C*; *Kroschwald et al., 2017*). We found that 1,6-hexanediol treatment of Pfk2-GFP wild-type cells during growth in hypoxia did not affect nascent G body formation (*Figure 7—figure supplement 1D*), indicating that 1,6-hexanediol cannot prevent nucleation of G bodies.

To test if a mutant with decreased G body formation would be more susceptible to disruption by 1,6-hexanediol, we used *snf1Δ* cells. We previously demonstrated that cells lacking Snf1, the yeast ortholog of AMP-activated protein kinase, had decreased G body formation and frequently formed multiple smaller foci per cell, rather than the single foci observed in wild type cells (*Jin et al., 2017*). Unexpectedly, 1,6-hexanediol treatment of *snf1Δ* cells increased the frequency of cells with a single Pfk2-GFP labeled G body (*Figure 7—figure supplement 1D*), such that they appeared superficially wild type. Therefore, 1,6-hexanediol promotes Pfk2-GFP aggregation in *snf1Δ* cells subjected to hypoxia, although this mode of aggregation may differ from normal G body formation. Similar results have been reported with extended treatments of high concentrations of 1,6-hexanediol for yeast P-bodies (*Wheeler et al., 2016*). Together, these findings suggest that G bodies are largely insensitive to 1,6-hexanediol and thus similar in physical properties to yeast stress granules.

Results from experiments with 1,6-hexanediol may not fully explain the physical properties of a granule. If G bodies behave as liquids, they would be expected to fuse and display exchange with the cytoplasm as has been demonstrated for a variety of granules such as P bodies (*Kroschwald et al., 2015*). To test if G bodies can exchange with the cytoplasm, we used fluorescence recovery after photobleaching (FRAP) to measure recovery kinetics of several G body-resident proteins fused to GFP (Pfk2, Fba1, and Eno2). Due to their small size, we photobleached whole G bodies and measured their recovery times. Despite much of the fluorescence in the cell residing in G bodies, we were able to detect some recovery of Pfk2-GFP in yeast grown in YPD as well as in synthetic media (*Figure 7—figure supplement 2A*). However, there was very weak recovery (9.5% in YPD and 15% in SMD) and the half time of recovery was on the order of minutes. Fba1-GFP had substantially more recovery (28%) but with a similar half time of recovery (25 min) (*Figure 7—figure supplement 2B*). Eno2-GFP also displayed recovery, but with dynamics that appeared linear and did not fit to an exponential model (*Figure 7—figure supplement 2C*). To test whether RNA affects G body dynamics, we used cells with a plasmid expressing Pfk2$^{D348S}$-MqsR-Flag and cells with a vector control grown in SMD-uracil to maintain the plasmid. The Pfk2-D348S variant has reduced enzymatic activity and its overexpression would therefore mitigate any effect of increased Pfk2 activity

(**Arvanitidis and Heinisch, 1994**). We treated the cells with a small dose of $CuSO_4$ (5 μM) and performed FRAP. Surprisingly, we detected slightly greater overall percentage of recovery (13% vs. 4.1%) for cells expressing Pfk2$^{D348S}$-MqsR-Flag, although the weak recovery was more rapid (5.7 min vs. 10.6 min) (**Figure 7—figure supplement 2D**). Taken together, when RNA is slightly depleted from G bodies by targeted RNase treatment, there is a modest enhancement in recovery of Pfk2-GFP, indicating that Pfk2 is more stably associated with G bodies in the absence of RNase. Thus, RNA may promote stronger binding of Pfk2-GFP in G bodies.

Consistent with fusion events in G body biogenesis, yeast cells display multiple Pfk2-GFP foci at early time points in hypoxia but only a single focus at later timepoints (**Jin et al., 2017**). To test directly G body fusion in vivo, we mated **a** and **α** yeast cells, each expressing Pfk2-GFP, and imaged the mating cells over time following hypoxic incubation. When cells were cultured in hypoxia for 18 hr and subsequently placed under a coverslip, we could track individual G bodies over several hours (**Figure 7—figure supplement 3A**). G bodies moved throughout the cell (**Figure 7A,B**; **Figure 7—video 1**, **Figure 7—video 2**). When a G body from one mating cell type met a G body from the other mating cell type, they were able to fuse into a single G body in the mated diploid cell. However, unlike granules in liquid-like states, G body fusion occurred over longer timescales: from as short as 2 min to as long as 60 min, with a median of 18 min from initial contact to fusion to form a single G body (**Figure 7A**). Although some G bodies stayed oblong in shape when fused, others became more spherical over tens of minutes, suggesting that they behaved as gels with fusion over long timescales. In accordance with these results, we also identified some cells that formed G bodies de novo during imaging. Initially, two distinct foci appeared from a diffuse cytoplasm, but over 60 min, these foci became brighter and fused within 2 min of initial contact (**Figure 7B**).

To gain a better understanding of the frequency of fusion events, we mated **a** cells expressing Pfk2-GFP to **α** cells expressing Pfk2-Azurite following hypoxic treatment and allowed them to settle in 24-well plates. Over successive timepoints (3, 5, 7 and 24 hr), we measured the fraction of foci with each label that were overlapping, adjacent to, or distinct from puncta with the other label (**Figure 7C**). These cells would be in a hypoxic environment allowing for some de novo granule formation. We observed a high frequency of overlapping puncta, which increased over time. Overlapping puncta likely arose from fusion, although we cannot rule out that a subset arose from de novo granule formation in mating cells. The fraction of mating cells with adjacent and unassociated puncta decreased from 41% after 3 hr mating to only 11% after 24 hr mating, whereas the fraction with overlapping or fused puncta increased (**Figure 7D,E**, **Figure 7—figure supplement 3B**). The progression from many particles to a single particle is consistent with fusion of granules. Taken together, these results suggest that G bodies can fuse over minutes and are largely insensitive to 1,6-hexanediol, properties reminiscent of gels.

## G body formation improves competitive fitness in yeast in hypoxia

To determine if RNA-mediated G body formation is important for cell survival or proliferation in hypoxia, we used a growth competition assay. By inoculating hypoxic cultures with log phase cells expressing either Pfk2-MqsR-Flag or a vector control and measuring G body formation, we could determine the expected fraction of cells with G bodies from a culture inoculated with a 1:1 mix of each population. An increase in the fraction of cells with G bodies from the 1:1 mix of each population relative to the expected fraction would indicate a competitive advantage for the cells from the vector control group (which form G bodies). Conversely, a decrease would indicate a competitive advantage for cells expressing the Pfk2-MqsR-Flag plasmid (which do not form G bodies) (**Figure 7—figure supplement 4A**). By comparing G body formation in the mixed population to the mean of each single population alone, we could determine the precise fraction of the total population arising from each population seeded. Strikingly, after 6 to 7 generations in hypoxia, we observed that vector control cells made up 65% of the culture when grown with Pfk2-MqsR-Flag expressing cells (**Figure 7—figure supplement 3B**). Since each population had been grown to log phase in normoxia, this difference was solely due to competitive growth in hypoxia. The advantage in vector control cells was not due to extra Pfk2 catalytic activity as cells expressing Pfk2$^{D348S}$-MqsR-Flag also had a disadvantage compared to vector control cells. Furthermore, cells expressing MqsR alone also had a growth advantage over both Pfk2-MqsR-Flag and Pfk2$^{D348S}$-MqsR-Flag expressing cells. Cells expressing Pfk2-MqsR-Flag-MqsA had only a small and not statistically significant increase in competitive fitness (**Figure 7—figure supplement 3B**). To determine if the competitive advantage

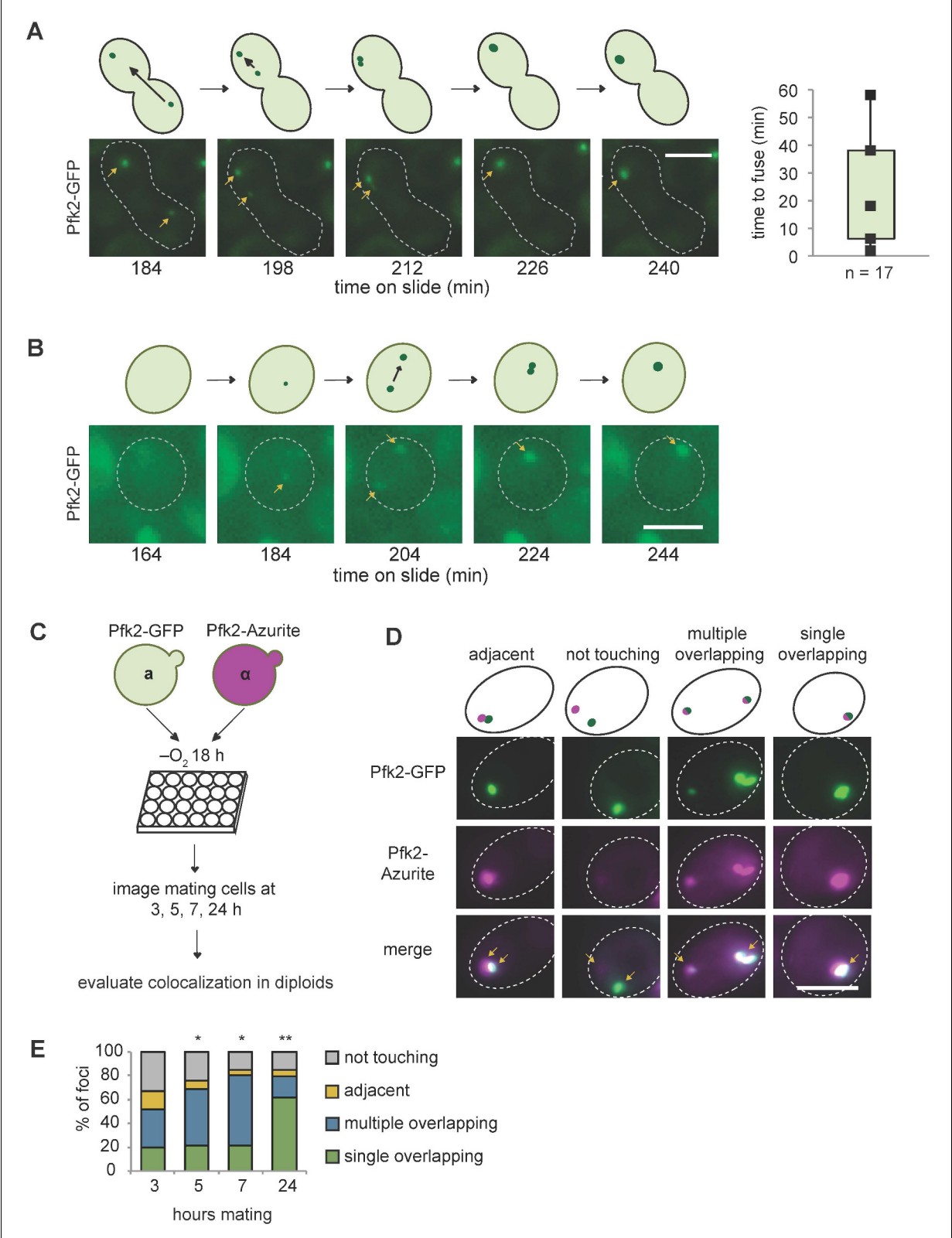

**Figure 7.** G bodies fuse in vivo. (**A–B**) Still images from time-lapse imaging of Pfk2-GFP in mating cells with cartoon. **a** and **α** cells each expressing Pfk2-GFP were cultured together in hypoxia for 18 hr, mounted on a slide, and imaged every 2 min for 3–4 hr. (**A**) G body fusion. Initially, foci are present in opposite ends of mating cells. One G body crosses to the other end and gradually fuses with the other G body. (Right) Quantification of time to fuse for multiple G body fusion events from initial contact to fusion. (**B**) De novo G body formation. Initially, Pfk2-GFP is diffuse in the highlighted cell. Two

*Figure 7 continued on next page*

*Figure 7 continued*

small puncta appear and gradually fuse into one larger focus, becoming more intense over time. (C) Mating for quantification of G body fusion. **a** and **α** cells expressing Pfk2-GFP and Pfk2-Azurite, respectively, are grown in hypoxia together and allowed to settle in 24-well plates. Cultures are sampled at 3, 5, 7 and 24 hr and phenotypes are evaluated. (D) Representative images of mating Pfk2-Azurite and Pfk2-GFP cells displaying different colocalization phenotypes. (E) Quantification of G body fusion at each time point. Foci were classified as single foci in cells that were overlapping, multiple overlapping foci in cells, adjacent foci in cells, or foci not touching or associated in cells. Data represent mean of three independent experiments (n > 40 foci per timepoint per replicate). Arrows indicate G bodies. All scale bars are 5 μm. *p*-values represent largest Chi square test of difference between subsequent timepoints for a replicate. $*p<10^{-5}$, $**p<10^{-10}$.

The online version of this article includes the following video, source data, and figure supplement(s) for figure 7:

**Source data 1.** Source data for *Figure 7A*.
**Source data 2.** Source data for *Figure 7E*.
**Figure supplement 1.** 1,6-Hexanediol partially dissolves G bodies but does not inhibit their formation.
**Figure supplement 1—source data 1.** Source data for *Figure 7—figure supplement 1A*.
**Figure supplement 1—source data 2.** Source data for *Figure 7—figure supplement 1B*.
**Figure supplement 1—source data 3.** Source data for *Figure 7—figure supplement 1C*.
**Figure supplement 1—source data 4.** Source data for *Figure 7—figure supplement 1D*.
**Figure supplement 2.** G body components exchange with the cytoplasm.
**Figure supplement 2—source data 1.** Source data for *Figure 7—figure supplement 2*.
**Figure supplement 3.** Schematics of mating experiments.
**Figure supplement 4.** G body formation confers a selective advantage to cells in hypoxia.
**Figure supplement 4—source data 1.** Source data for *Figure 7—figure supplement 4B*.
**Figure supplement 4—source data 2.** Source data for *Figure 7—figure supplement 4C*.
**Figure 7—video 1.** Fusion of G bodies.
https://elifesciences.org/articles/48480#fig7video1
**Figure 7—video 2.** Fusion of G bodies.
https://elifesciences.org/articles/48480#fig7video2

increases with the number of generations, we made serial dilutions of the 1:1 mixtures of vector control cells and Pfk2-MqsR-Flag cells, and each population alone, and allowed them to grow for 24 hr and 48 hr and plotted the fraction of cells from the vector control population against the number of generations determined by the increase in OD. To determine if there was a bias in seeding the cells, we also measured colony forming units per ml of cells from each population of log phase cells. We observed an increase in the population of cells with the vector control plasmid with increasing numbers of generations (*Figure 7—figure supplement 3C*), consistent with a significant advantage for cells with G bodies. Thus, specifically depleting cells of G bodies by degrading RNA puts them at a competitive disadvantage to cells capable of forming G bodies.

## Discussion

### Physical properties of G bodies

Here we report that G bodies have properties of phase separated gels in vivo. First, similar to yeast and mammalian stress granules, G bodies lack membranes and contain chaperones as well as proteins with intrinsically disordered regions (*Jain et al., 2016*; *Jin et al., 2017*). Second, we found that G bodies were much less sensitive to 1,6-hexanediol than liquid phase-separated structures like P bodies (*Figure 7—figure supplement 1*), extending their similarities to stress granules. However, their size appears reduced after 1,6-hexanediol treatment (*Figure 7—figure supplement 1*), indicating partial dissolution of the granules. This may be reminiscent of the 'dynamic shell' model proposed for stress granules in which a fluid shell surrounds a solid core structure (*Jain et al., 2016*), although there is currently no direct evidence of substructure within G bodies. Third, while G bodies can fuse in vivo (*Figure 7*), a property of liquids (*Boeynaems et al., 2018*; *Elbaum-Garfinkle et al., 2015*; *Hyman et al., 2014*), this fusion takes place on the order of tens of minutes, unlike fusion of P bodies, which takes only seconds (*Kroschwald et al., 2015*). Similar to P bodies, P granules in the *C. elegans* germline and stress granules can undergo fission and fusion (*Brangwynne et al., 2009*; *Kedersha et al., 2005*; *Ohshima et al., 2015*). The slow fusion dynamics of G bodies are consistent with the weak, but measurable exchange with the cytoplasm observed by FRAP of whole G bodies

(*Figure 7—figure supplement 2*). Indeed, formation of G bodies when cells are transitioned to hypoxia involves fusion of smaller granules into larger structures. Early fusion events were insensitive to 1,6-hexanediol due to the presence of single foci even after extended 1,6-hexanediol treatment during hypoxia (*Figure 7—figure supplement 1*). Additionally, even after extended periods in hypoxia, G bodies fuse in mating cells, demonstrating that 'mature' G bodies can still fuse (*Figure 7*). Fourth, in contrast to other RNP granules, G bodies are remarkably stable. Like stress granule cores, which can be isolated and are stable in a lysate, G bodies can be purified intact (*Jin et al., 2017*). However, unlike stress granules, which can disperse on the order of tens of minutes following removal of stress, and nucleoli, which rapidly assemble and disassemble during mitosis (*Brangwynne et al., 2011*; *Feric et al., 2016*; *Jain et al., 2016*; *Tsai et al., 2008*; *Walters et al., 2015*), G bodies can persist for tens of hours as revealed in our experiments inducing Pfk2-MqsR-Flag following hypoxic incubation (*Figure 5*). One possibility is that G bodies initially form from a liquid state and gradually solidify, becoming more similar to protein aggregates of FUS and IDRs of other proteins over time (*Lin et al., 2015*; *Mateju et al., 2017*; *Murakami et al., 2015*). However, G bodies, once formed, can be disrupted by the targeting of an RNase, demonstrating that they are not static. The effect of Pfk2-MqsR-Flag induced after hypoxic treatment (when G bodies have already formed) was largely independent of the concentration of $CuSO_4$ added (*Figure 5*). Thus, a small amount of RNA degradation was sufficient for G body fragmentation. The fragmented foci induced after hypoxic treatment appeared larger than single puncta in vector control cells (*Figure 5*), suggesting additional Pfk2-GFP could accumulate in these puncta in the absence of RNA. This is consistent with RNA primarily being required for G body fusion and nucleation. It is unclear whether resident G body proteins ever return to the soluble cytosolic pool when cells are shifted back from hypoxic to normoxic conditions.

Similar to proteins in other granules, protein recruitment to G bodies relies on multivalent interactions. Both the N- and C-terminal domains of Pfk2 are necessary but not sufficient for G body recruitment (*Figure 6*). Pfk1 is recruited to G bodies via an interaction with Pfk2 (*Figure 6*). Additionally, protein-RNA interactions are required for G body formation (*Figure 4*). Such multivalent interactions are necessary for phase separation and can drive granule formation (*Hyman et al., 2014*; *Li et al., 2012*).

## RNA in phase separation

By targeting destructive RNase fusion proteins to sites of G body formation, like a Trojan Horse, we show that RNA is required for G body formation in vivo (*Figure 4*). Degradation of RNA in existing G bodies led to multiple puncta, suggesting that RNA is required for G body stability or integrity (*Figure 5*). Furthermore, FRAP data indicated that Pfk2-GFP in G bodies targeted by a weak induction of RNase was slightly more dynamic than untreated G bodies, suggesting that RNA contributes to Pfk2 recruitment to G bodies (*Figure 7—figure supplement 2*). Our mass spectrometry analysis of the 'RBPome' identified hundreds of mRBPs, including glycolysis enzymes, consistent with and adding to previous work on RPBome discovery (*Figure 1*; *Matia-González et al., 2015*; *Beckmann et al., 2015*; *Scherrer et al., 2010*). By PAR-CLIP-seq, a comprehensive picture of an extensive glycolysis enzyme-bound transcriptome is emerging (*Figure 2*; *Shchepachev et al., 2019*). Intriguingly, these enzymes bind similar transcripts primarily in the 3' UTRs and coding regions of their substrate mRNAs (*Figure 2*). We identified a common set of transcripts bound by multiple glycolysis enzymes in normoxic conditions (*Figure 2*). We determined, by RIP-qPCR and RIP-seq of purified G bodies, that many of these mRNA substrates were then recruited along with their protein binding partners to G bodies in hypoxia (*Figure 3*). smFISH of G body-associated RNAs revealed the extent to which RNA colocalization with G bodies correlated with the degree of enrichment (*Figure 3*). Surprisingly, no RNA analyzed by smFISH was strongly enriched in G bodies, although some RNAs partitioned up to 2-fold more in G bodies than others. In contrast, stress granules can concentrate up to 95% of particular mRNA species (*Khong et al., 2017*), although this is within multiple granules. Therefore, while RNAs are required for G body formation, there may be only modest preference for which specific RNA species are recruited to G bodies, as long as they bind glycolysis enzymes.

We propose a model in which RNA serves as a scaffold for G body nucleation and growth. Consistent with this model, addition of RNA promotes the aggregation of the glycolysis enzyme, Cdc19, in vitro (*Saad et al., 2017*). Among the commonly bound transcripts were a number of mRNAs

encoding the glycolysis enzymes themselves. Weak, dynamic interactions between G body components with RNA and with each other would allow for the growth of G bodies. It has been proposed that RNA interactions with glycolysis enzymes can facilitate post-transcriptional regulation of the pathway (*Matia-González et al., 2015*). This mode of regulation could additionally contribute to G body formation through concentration of nascent proteins due to spatially segregated translation of glycolysis enzyme mRNAs. Interactions with glycolysis enzyme mRNAs may specifically facilitate multivalent interactions of glycolysis enzymes with RNA by providing a common set of substrates. Alternatively, hypoxia may weaken interactions of glycolysis enzymes with RNAs, allowing for G bodies to form via protein-protein interactions. Many RNAs are localized peripherally to G bodies (*Figure 3*). These RNAs may then allow G bodies to fuse by binding the glycolysis enzymes in other G bodies, explaining the loss of foci with loss of RNA. Experiments measuring binding affinity of glycolysis enzymes to mRNA in hypoxia and normoxia will be required to differentiate these possibilities.

## Phase separation in the control of metabolic pathways

Although spatial organization of pathways to concentrate constituent enzymes is not novel, phase separation is emerging as a new mechanism to achieve this type of organization. Even for glycolysis, pathway compartmentalization is known to occur; glycolysis enzymes are concentrated in membrane-bound compartments called glycosomes in various protozoa, including in *Trypanosoma brucei*, which can survive in anaerobic conditions (*Michels et al., 2006*; *Opperdoes, 1987*). However, phase separation is mechanistically distinct in that it does not require formation of a membrane or specific transporters. G bodies represent an addition to the known metabolic pathways organized by phase separation mechanisms. Three recent studies have demonstrated glycolysis enzyme coalescence in hypoxia, which is associated with increased rates of glucose flux (*Jang et al., 2016*; *Jin et al., 2017*; *Miura et al., 2013*). Thus, G bodies and glycosomes may represent a case of convergent evolution. Similar structures also form in the neurons of *C. elegans*, where enzyme clustering in response to hypoxia was associated with proper synaptic function, suggesting that glycolysis enzymes coalesce to increase glycolysis and meet the local high energy demand during synapses (*Jang et al., 2016*). Furthermore, cancer cell lines form small aggregates of glycolysis enzymes even in the presence of oxygen, suggesting that concentrating glycolysis enzymes is a highly conserved process (*Kohnhorst et al., 2017*). Additionally, pyrenoids, which enhance the rate of carbon fixation in *Chlamydomonas* by concentrating $CO_2$ in the presence of ribulose 1,5 bisphosphate carboxylase oxygenase (RuBisCo), were shown to form via phase separation (*Freeman Rosenzweig et al., 2017*). Carbon metabolism, then, can be controlled both at the level of fixation and harvesting via phase separation into distinct compartments. Here, we demonstrate that cells capable of forming G bodies possess a distinct growth advantage over multiple generations when co-cultured with cells that cannot form G bodies (*Figure 7—figure supplement 3*). Thus, RNA induced coalescence of glycolysis enzymes promotes proliferation in hypoxia, likely by enhancing rates of glycolysis.

The precise mechanism of enhanced glycolysis activity is unclear in G bodies. Recent in vitro measurements of dextranase activity in artificial liquid-liquid phase separated compartments suggest that phase separation can enhance reaction rates through decreased substrate inhibition (*Kojima and Takayama, 2018*). The G body resident factor phosphofructokinase mediates one of the irreversible steps in glycolysis. It is also subject to substrate inhibition by ATP and may experience increased specific activity akin to release from substrate inhibition when dextranase undergoes liquid-liquid phase separation. However, purinosomes, which are complexes composed of purine biosynthesis enzymes, are thought to enhance pathway activity through substrate channeling (*An et al., 2008*; *Zhao et al., 2013*). Concentrating enzymes in RNP granules may achieve similar results or enhance reaction flux rates via concentrating intermediate metabolites. Alternatively, concentration of energy producing enzymes with their cognate mRNAs may facilitate enhanced translation of the glycolysis enzymes, thus promoting pathway activity without increasing specific activity of the enzymes. Understanding the mechanism and function of metabolic pathway enhancement by phase separation will require future mechanistic studies.

## Materials and methods

### Yeast strains and culturing

Strains used are listed in *Supplementary file 3*. Deletion mutants and integrated transgenic strains were generated through transformation of amplified auxotrophic markers at the site of PCR amplified fragments with at least 40 nt of overlapping sequence to 3' and 5' UTR sequences. GFP mutants were derived from the yeast GFP tagged library (*Huh et al., 2003*). TAP-tagged mutant strains were derived from the yeast TAP library (*Ghaemmaghami et al., 2003*). Plasmids were transformed with standard lithium acetate transformation (*Gietz and Woods, 2002*) and selected on SD with appropriate amino acid supplements and auxotrophic selection.

Cells were grown as indicated either in YPD (2% peptone, 1% yeast extract, 5% glucose) or SMD (0.67% yeast nitrogen base, 5% glucose, appropriate amino acid and vitamin supplements). For hypoxic growth of yeast for imaging, cells were reinoculated from stationary phase starter cultures in 0.5–1 ml of the indicated media in 24-well plates at an $OD_{600}$ of 0.05. Cells were grown for the indicated amounts of time in hypoxia using the AnaeroPack system in 2.5 L boxes (Mitsubishi Gas Chemical). For larger-scale biochemical experiments, cells were grown in 125 ml Erlenmeyer flasks in 10–25 ml of media in 7 L boxes. For analysis of copper inducible proteins, cells were grown in SMD lacking uracil and supplemented with $CuSO_4$ at varying concentrations.

### Glucose starvation and 1,6-hexanediol treatment

For glucose starvation, cells were reinoculated from YPD starter cultures into SMD at an $OD_{600}$ of 0.15. Cells were grown for 6 hr shaking at 30°C to log phase and pelleted for 2 min at 500 x g. Cells were washed and resuspended in SM media lacking glucose with or without 5% 1,6-hexanediol for 30 min at room temperature. Cells were either immediately imaged or had 1,6-hexanediol in PBST directly added to the media, incubated for 1 hr at room temperature, and then imaged.

For hypoxic treatment with 1,6-hexanediol, cells were grown in YPD and reinoculated in SMD with 0%, 1%, or 2% 1,6-hexanediol at an $OD_{600}$ of 0.05 and grown 20 hr in hypoxia and imaged. For treatment with 1,6-hexanediol after hypoxia, cells were cultured in the same way and had 1,6-hexanediol in PBST directly added to the media, incubated for 1 hr at room temperature, and then imaged.

### Plasmid construction

Plasmids were generated by overlap extension PCR fusing fragments amplified from genomic DNA to fragments amplified from plasmid sources. Pfk2-Rnase A was generated by fusing the Pfk2-linker sequence from THY62 to RNase A, omitting the signal peptide from the pET22B RNase A plasmid, which was a gift from Ronald Raines (Addgene plasmid #58903). Pfk2-MqsR was generated by fusing the Pfk2-linker sequence from THY62 to MqsR amplified with a 3' 1X Flag peptide sequence from the pSLC-241 plasmid, which was a gift from Swaine Chen (Addgene plasmid #73194). Pfk2-MqsR-MqsA was generated by fusing Pfk2-MqsR-Fl to MqsA and subsequently fusing the Pfk2 3' UTR. Cytoplasmic MqsR was generated by amplifying MqsR from pSLC-241. Each construct was subsequently fused to the Pfk2 3' UTR amplified from genomic DNA and subcloned into pCu416CUP1 (ATCC 87729) using Spe1 and Xho1 sites. Pfk2-Rnase A$^{H12A}$ was introduced with site-directed mutagenesis and verified by sequencing. Pfk2$^{D348S}$ was introduced using site directed mutagenesis and verified by sequencing.

### Yeast immunofluorescence

Yeast immunofluorescence protocol was adapted from *Amberg et al. (2005)*. Briefly, cells were grown in hypoxia for indicated periods of time to an $OD_{600}$ of ~2 in 1 ml volumes. Formaldehyde was added to culture media to 1% for 10 min and then washed away. Cells were resuspended in KM (50 mM potassium phosphate, 5 mM $MgCl_2$, pH 6.5) with 4% formaldehyde for 1 hr at 30°C and washed twice with KM and once with KM with 50% sorbitol (KMS). Cells were resuspended in 50 µL KMS with 5 U Zymolyase (Zymo E1004) for 20 min at 37°C. Cells were washed and resuspended in KMS and adhered to Superfrost Plus (Fisher 12-550-15) slides. For probing Pfk2-RNase A, cells were dipped in ice-cold methanol for 6 min followed by acetone for 30 s and dried at 50°C. For Pfk2-MqsR, to preserve GFP fluorescence, this step was omitted. Cells were blocked in 1% BSA in PBST 2

hr at room temperature. For Pfk2-RNase A, cells were incubated with 1:200 rabbit anti-S tag antibody (Genscript A00625) overnight and 1:1000 mouse anti-GFP antibody (Thermo Fisher A-11120) for Pfk2-RNase A and 1:200 mouse anti-Flag antibody in PBST overnight at 4°C. Cells were washed three times in PBST and 1:500 donkey anti-mouse Alexa Fluor 488 antibody (A-21202), and 1:500 goat anti-rabbit Alexa Fluor 647 antibody (A-21245) for Pfk2-RNase A in PBS was added for 1 hr at room temperature. For Pfk2-MqsR-Fl experiments, 1:500 donkey anti-mouse Alexa Fluor 647 antibody (A-31571) was added in PBST for 1 hr at room temperature. Cells were mounted in Vectashield plus DAPI (Vector laboratories, H1200) and imaged.

## Yeast fluorescence microscopy imaging and analysis

All cells were imaged using a Zeiss AxioImager M2 with an ORCA-Flash 4.0 LT camera with band pass GFP filter (Zeiss 38 HE eGFP) illuminated with a 488 nm LED with either 40X or 100X objective taking Z stacks to cover the entire cell. Immunofluorescence images were taken with illumination from a mercury halide arc lamp and band pass GFP filter and CY5 filter (Zeiss 50).

For assaying G body formation, cells were manually counted and classified into one of three categories: cells with single puncta, cells with multiple puncta, and cells with no puncta. At least 100 cells were considered for each replicate and condition.

For mating experiments, cells of each mating type were mixed at an $OD_{600}$ of 0.05 and grown 18 hr -$O_2$. Cells were allowed to settle in 24-well plates for mating to proceed and sampled after 3, 5, 7, and 24 hr. For mating cells with Pfk2-Azurite and Pfk2-GFP, Azurite was imaged with a BFP band pass filterset (Zeiss 96 HE) illuminated with a 365 nm LED, and GFP was imaged as above. To compensate for high background, mating cells were resuspended in PBS before imaging. Z stacks were taken in each channel and brightfield. Overlapping puncta were manually counted in all Z planes, and puncta were classed into four categories: overlapping puncta in cells with one focus, overlapping puncta in cells with multiple foci, adjacent puncta, and puncta not associated with other puncta. For mating cells with Pfk2-GFP only to observe kinetics, cells were mixed and grown 18 hr -$O_2$ and placed on a slide. Fields of cells were imaged taking Z stacks every 2 min through a GFP filter.

For size distributions of G bodies, local maxima were identified in maximum intensity projections in FIJI (*Schindelin et al., 2012*) using the Find Maxima tool. A square with sides of 4.9 μm was drawn around each maximum. Average background from three separate spots in each image was subtracted. Using custom Python scripts (*Supplementary file 5*), each focus was fit using nonlinear least squares to a 2-dimensional Gaussian distribution of the form:

$$f(x,y) = b + A*e^{-((\frac{cos^2\theta}{2\sigma_x^2} + \frac{sin^2\theta}{2\sigma_y^2})(x-x_0)^2 + 2(-\frac{sin2\theta}{4\sigma_x^2} + \frac{sin2\theta}{4\sigma_y^2})(x-x_0)(y-y_0) + (\frac{sin^2\theta}{2\sigma_x^2} + \frac{cos^2\theta}{2\sigma_y^2})(y-y_0)^2))}$$

where b is the baseline fluorescence, A is the amplitude of the peak fluorescence of the focus, $x_0$ and $y_0$ define the center coordinate, $\sigma_x$ and $\sigma_y$ represent the standard deviation along each axis and θ defines the rotation of the punctum.

## Yeast PAR-CL-Mass spectrometry

3 L of BY4742 were cultured from an initial $OD_{600}$ of 0.003 and grown until $OD_{600}$ of 0.7–0.8. Cycloheximide was added to a final concentration of 0.1 mg/ml and incubated at 30°C for 5 min. Cells were pelleted by centrifugation for 5 min at 4,000 rpm at 18°C using a JLA-10.5 rotor in an Avanti J-26XP centrifuge, resuspended in 10 ml 1X PBS (with 0.1 mg/ml cycloheximide), transferred to a 150 mm glass Petri dish, placed on ice, and irradiated 4 times with 365 nm UV light at 150 mJ/cm$^2$ using a UVP CL-1000L UV crosslinker. The cells were then transferred to a 15 ml conical tube and pelleted for 3 min at 3000 x g at room temperature. After removing the PBS, the cells were frozen in liquid nitrogen. For the negative control, cells were frozen without UV irradiation. Frozen cell pellets were pulverized for two cycles, each for 1 min, at 30 Hz on a Retsch MM 4000 ball mill homogenizer. Sample chambers were pre-chilled in liquid nitrogen and re-chilled between cycles. The resulting frozen powdered homogenate was resuspended in 3 ml of polysome lysis buffer (20 mM HEPES, pH 7.5, 140 mM KCl, 1.5 mM MgCl2, 1% Triton X-100, 1X cOmplete Mini Protease Inhibitor, EDTA free, 0.1 mg/ml cycloheximide) and incubated on ice for 10 min. Cell debris was removed by centrifugation for 2 min at 3000 x g at 4°C. The supernatant fraction was clarified by a 20,000 x g spin for 10 min at 4°C and supplemented with 12 μl SUPERase·In (20 U/μl). 0.3 ml of 1 M (34.2% w/v) sucrose cushion solution, prepared in polysome lysis buffer, was layered on the bottom of each 11 × 34 mm

polycarbonate centrifugation tube. 3 ml of clarified lysate was loaded onto three sucrose cushions (1 ml per cushion) and spun for 80 min at 54,000 rpm at 4℃ in a TLS-55 rotor using an Optima Ultra-centrifuge MAX-E. After centrifugation, the top 1 ml of solution was recovered from each tube (3 ml in total), mixed with 1.5 g of guanidine thiocyanate (GuSCN, Promega V2791), vortexed to dissolve GuSCN, and heated for 5 min at 65℃. A Zeba desalting column (7K MWCO, 10 ml, Pierce 89894) was used to remove GuSCN and to exchange buffer to 50 mM NaCl buffer (20 mM HEPES, pH 7.3, 50 mM NaCl, 0.5% Sarkosyl, 1 mM EDTA). Buffer-exchanged lysate was combined with 0.1 vol of 5 M NaCl to adjust the salt concentration to 0.5 M and supplemented with 6 µl SUPERase·In (20 U/µl). The lysate was incubated with 37.5 mg of oligo(dT)25 beads (NEB S1419S) for 30 min at 4℃ on a Nutator. Beads were washed four times with ice-cold low-salt wash buffer (20 mM HEPES, pH 7.3, 0.2 M NaCl, 0.2% Sarkosyl, 1 mM EDTA). RNAs were eluted in 1 ml of elution buffer (10 mM HEPES, pH 7.3, 1 mM EDTA) by heating for 3 min at 65℃. Eluted RNAs were concentrated to 80 µl with Amicon spin filters (3 KD cutoff, Millipore UFC500324). Concentrated RNAs were mixed with 40 µl of 3X SDS sample buffer (150 mM Tris, pH 6.8, 6% SDS, 30% glycerol, 3% beta-mercaptoethanol, 37.5 mM EDTA, 0.06% Bromophenol blue) and heated for 5 min at 65℃. 20 µl of sample was loaded per lane (six lanes total) of a 4–12% NU-PAGE Bis-Tris gel and run for 10 min at 100 V, followed by 70 min at 150 V. The gel was stained with Colloidal Blue (Invitrogen LC6025), and a gel piece, 0.1 cm-1.0 cm below the well, was excised and stored at −80℃ before mass spectrometry analysis.

For mass spectrometry, unless otherwise noted, all chemicals were purchased from Thermo Fisher Scientific (Waltham, MA). Deionized water (18.2 MW, Barnstead, Dubuque, IA) was used for all preparations. Buffer A consists of 5% acetonitrile, 0.1% formic acid; buffer B consists of 80% acetonitrile, 0.1% formic acid; and buffer C consists of 500 mM ammonium acetate. All buffers were filtered through 0.2 mm membrane filters (PN4454, Pall Life Sciences, Port Washington, NY). In-gel digestion was performed as in *Jensen et al. (1999)* with the following adjustments: Gel particles were rehydrated with 10 mM Tris (2-carboxyethyl) phosphene in 100 mM NH4HCO3 and incubated for 30 min at room temperature. Digestion buffer was 50 mM NH4CO3, 5 mM CaCl$_2$, containing 12.5 ng/µl trypsin. The gel pieces were rehydrated at room temperature for 30–45 min. The enzyme supernatant fraction was not removed, and 50 µl digestion buffer, without enzyme, was added before overnight digestion. A MudPIT microcolumn (*Wolters et al., 2001*) was prepared by first creating a Kasil frit at one end of an undeactivated 250 mm ID/360 mm OD capillary (Agilent Technologies, Inc, Santa Clara, CA). The Kasil frit was prepared by briefly dipping a 20- to 30 cm capillary in well-mixed 300 ml Kasil 1624 (PQ Corporation, Malvern, PA) and 100 ml formamide, curing at 100℃ for 4 hr and cutting the frit to ~2 mm in length. Strong cation exchange particles (SCX Partisphere, 5 mm dia., 125 Å pores, Whatman) were packed in-house from particle slurries in methanol to 2.5 cm. 2.5 cm reverse phase particles (C18 Aqua, 3 mm dia., 125 Å pores, Phenomenex, Torrance, CA) and were then packed into the capillary using the same method as SCX loading, to create a biphasic column. The MudPIT microcolumn was equilibrated using 60% buffer A, 40% buffer B for 5 min, and followed by 100% buffer A for 15 min. An analytical RPLC column was generated by pulling a 100 mm ID/360 mm OD capillary (Polymicro Technologies, Inc, Phoenix, AZ) to 5 mm ID tip. Reverse phase particles (Aqua C18, 3 mm dia., 125 Å pores, Phenomenex, Torrance, CA) were packed directly into the pulled column at 800 psi until they were 12 cm long. The column was further packed, washed, and equilibrated with buffer B followed by buffer A. The MudPIT microcolumn was connected to an analytical column using a zero-dead volume union (Upchurch Scientific (IDEX Health and Science), P-720–01, Oak Harbor, WA). LC- MS/MS analysis was performed using an Eksigent nano-flow pump and a Thermo LTQ-Orbitrap using an in-house-built electrospray stage. MudPIT experiments were performed where each step corresponds to 0, 10, 20, 30, 40, 50, 60, 70, 80, 90, and 100% buffer, C being run for 5 min at the beginning of each gradient of buffer B. Electrospray was performed directly from the analytical column by applying the ESI voltage at a tee (150 mm ID, Upchurch Scientific) while flowing at 350 nl/min through the columns. Electrospray directly from the LC column was done at 2.5 kV with an inlet capillary temperature of 250℃. Data-dependent acquisition of MS/MS spectra with the LTQ-Orbitrap were performed with the following settings: MS/MS on the 10 most intense ions per precursor scan, one microscan, unassigned and charge state one reject; dynamic exclusion repeat count, 1, repeat duration, −30 s; exclusion list size 120; and exclusion duration, 120 s. Tandem mass spectra were extracted from raw files using RawExtract 1.9.9 (*McDonald et al., 2004*) and were searched against a yeast protein database (http://www.yeastgenome.org) with reversed sequences using ProLuCID (*Peng et al., 2003*; *Xu et al., 2015*). The search space included

all fully and half-tryptic peptide candidates. Carbamidomethylation (+57.02146) of cysteine was considered a static modification. Peptide candidates were filtered using DTASelect (v2), with the following parameters: `-p 1 - y 1 –trypstat –fp 0.01 –extra -DM 10 -DB –dm –in` (*McDonald et al., 2004*; *Tabb et al., 2002*).

## Yeast PAR-CLIP western blotting and autoradiography

RBP validation was performed similar to the PAR-CLIP protocol but omitting the linker ligation steps. Cells collected from 50 ml of log-phase culture were used in each IP. After autoradiography, the same membrane was blotted with a peroxidase anti-peroxidase soluble complex antibody (Sigma-Aldrich P2416) at 1:10,000 dilution and developed with ECL substrates (Pierce 32209).

## Yeast PAR-CLIP-seq (Pfk2, Eno1, Fba1)

PAR-CLIP was performed as described previously (*Freeberg et al., 2013*). Briefly, yeast were grown to mid-log phase and irradiated with 365 nm UV. Cross-linked cells were lysed, treated with Rnase T1, and mixed with IgG magnetic beads to affinity isolate each TAP-tagged protein. Lysates were then subjected to RNase T1 digestion, CIP treatment, 3' DNA linker ligation, 5' end phosphorylation, and SDS-PAGE. After nitrocellulose transfer, cross-linked RNAs were visualized by autoradiography. Bands corresponding to each protein were excised and incubated with proteinase K. RNAs were collected by centrifugation and loaded onto a 6% TBE UREA gel. Gel pieces corresponding to 70–90 nt RNA were excised followed by amplification of the RNA fragments by RT-PCR. Amplicons were purified, run on a 10% TBE gel, and gel pieces corresponding to 96–116 bp DNA were excised. DNA fragments were amplified by PCR for two rounds and sequenced on an Illumina HiSeq 2000 sequencer. All primers used are as listed previously (*Freeberg et al., 2013*). Specifically, Indexes 1, 3, and 1 (for Pfk2, Fba1, and Eno1, respectively) barcoded 3' DNA linker oligos and reverse transcription primers were used.

Sequenced PAR-CLIP-seq read data were processed as described for Puf3 PAR-CLIP-seq (*Freeberg et al., 2013*). Briefly, reads were processed to remove linkers and sorted into libraries based on 6-nt barcodes. Next, reads were removed if they met any of the following criteria:<18 nt, only homopolymer As, missing 3' adapter, 5'–3' adapter ligation products, 5'–5' adapter ligation products, and low quality (more than four bases with quality scores below 10 or more than six bases with a quality score below 13). High-quality reads were mapped to the *S. cerevisiae* genome (S288C, sacCer3) with Bowtie (*Langmead, 2010*) using the following parameters: -v 3 (map with up to three mismatches), -k 275 (map at up to 275 loci), `–best`, and –strata.

## Pfk2, Fba1, and Eno1 binding site generation

Reads were assembled into binding sites by aggregating overlapping reads harboring 0–2 T-to-C conversion events. Only binding sites containing at least 1 T-to-C conversion event were considered high-confidence binding sites. For each library, the counts of sequencing reads covering each position within a binding site were averaged and normalized to the total number of millions of mapped reads in that library. To filter off low-coverage binding sites, a reads-per-million (RPM) mapped reads threshold for each library was empirically determined by simulating replicate data from each PAR-CLIP-seq dataset. Two sets of binding site RPM values were randomly sampled from all binding sites passing a minimum RPM threshold in a single dataset such that each sample contained 20% of the binding sites. A non-parametric two-sample Kolmogorov-Smirnov (K-S) test was performed on the two sets of RPM values, and the resulting K-S test statistic was recorded. This test was repeated 10,000 times for each of 36 RPM threshold values ranging from 0 to 25. Mean K-S test statistic values were plotted for each RPM threshold value, and a final binding site RPM threshold value for the library was chosen when the K-S test statistic stabilized (*Figure 1—figure supplement 1B*). For Pfk2, an empirical RPM threshold of 5 RPM was used, and a threshold of 0.5 RPM was used for Eno1 and Fba1. After filtering, binding site RPM values were normalized to gene expression RPKM values from previously published data (*Freeberg et al., 2013*). Binding sites were annotated using custom scripts to known genomic elements in the S288C (sacCer3) yeast genome. ORFs with unannotated UTRs were hierarchically assigned UTRs from the following: (*Nagalakshmi et al., 2008*; *Yassour et al., 2009*). GO term analysis was performed using the g:Profiler web server (*Reimand et al., 2016*).

## G body immunoprecipitation and RNA extraction

Protein G Dynabeads (ThermoFisher Scientific, 1004D) were conjugated to mouse anti-Flag M2 antibody (Sigma Aldrich F1804) as previously described (*Jin et al., 2017*). Beads were stored in 10% BSA in lysis buffer (150 mM NaCL, 25 mM Tris-HCl pH 7.5, 5 mM EDTA, 0.5 mM Dithiothreitol, 0.5% NP-40, 1 cOmplete mini EDTA-free protease inhibitor cocktail tab per 5 ml (Roche, 0463159001), 40 U RNaseOUT (Thermo Fisher 10777019) per ml) until use. Pfk2-GFP-Flag cells and BY4742 cells were reinocculated from overnight YPD starter cultures into 125 ml of YPD at an $OD_{600}$ of 0.005 split into five 125 ml Erlenmeyer flasks. Cells were grown in hypoxia for 18 hr. Cells were imaged to ensure normal G body formation. 100 $OD_{600}$ of cells were spun down at 3000 x g for 10 min and decanted with remaining media being aspirated. Cells were resuspended in 1.5 ml of lysis buffer and lysed by glass bead lysis for 10 min alternating vortexing and ice every 30 s. Large cell debris were removed from the supernatant by centrifugation at 500 x g for 5 min at 4°C. The supernatant was precleared once with 25 µl of Protein G Dynabeads nutating for 30 min at 4°C. G bodies were then pelleted by centrifugation at 5000 x g for 10 min and resuspended in 1 ml of lysis buffer and incubated with Dynabeads for 2 hr at 4°C. After 1 hr, Dynabeads were imaged to ensure capture of G bodies. 250 µl of the flow through was saved for RNA extraction. Beads were washed 3X with 1 ml of lysis buffer and once with 1 ml of Proteinase K buffer (100 mM Tris pH 7.5, 50 m NaCL, 10 mM EDTA). Beads were resuspended in 120 µl 2 mg/ml 3X Flag peptide (Millipore F4799) in Proteinase K buffer and eluted for 30 min at 37°C on a thermomixer at 850 rpm. After 15 min, 1 µl of the eluate was imaged to validate elution of G bodies from Dynabeads. The eluate was mixed 1:1 with 8 mg/ml Proteinase K (Thermo Fisher 25530015) in proteinase K buffer for 1 hr in a thermomixer at 37°C at 850 rpm. Eluates and flow-through RNA were extracted with Tri Reagent (Sigma-Aldrich T9424) according to the manufacturer instructions. The aqueous phase was mixed 1:1 with isopropanol with 1 µl of glycogen (Thermo Fisher R0561) and precipitated for 4 hr at −80°C. RNA was pelleted by centrifugation at 4°C at 21,000 x g for 30 min. RNA was washed 3 times with 80% ethanol and re-precipitated in 80% ethanol with sodium acetate overnight at −80° C. RNA was pelleted again and washed again and dissolved in 30 µl of $H_2O$. For each sample, 10 µl of RNA was used to generate two technical replicates of cDNA with the High-Capacity cDNA reverse transcription Kit (Thermo Fisher 4368813) per the manufacturer's instructions. Individual probes (*Supplementary file 2*) were used to measure different RNA species using Absolute Blue qPCR SYBR Green (Thermo Fisher AB4166B) per the manufacturer instructions with a Bio-RAD CF96 Real Time PCR thermal cycler. As the flow through RNA represented 25% of the total flow through, two cycles were subtracted from the resulting Cq values. For each probe, the ΔCq of elution RNA – flow through RNA was measured and the percent of input was plotted.

For two replicates, sequencing libraries of total RNA, flow through RNA and IP RNA were prepared with the NEBNext Ultra Directional RNA Library Prep Kit for Illumina (NEB, Beverly, MA, USA) with between 5 and 25 ng of RNA. Libraries were analyzed on an Agilent 4200 TapeStation. Libraries were sequenced using a HiSeq2500 in rapid run mode. Data were trimmed and analyzed using Galaxy (*Afgan et al., 2018*). Libraries were assessed using FastQC. Adapters were trimmed with TrimGalore and reads were mapped to the Ensembl assembly of the *S. cerevisiae* genome (R64-1-1.93) using Bowtie (*Langmead, 2010*). Reads were counted, and normalized to RPM within each library. Since BY4742 IPs (as measured by qPCR) recovered less than 10% of each G body IP RNA tested, data were only compared to RNA from Pfk2-GFP-Flag preps.

## smFISH and analysis

Custom Stellaris FISH probes labeled with Quasar 670 dye were obtained from LGC, BioSearch Technologies (SMF-1065–5, BioSearch Technologies, Petaluma, Ca). Cells were inoculated from overnight cultures and grown in hypoxia for 16 hr. Cells were fixed and cell walls were digested using Zymolyase. Cells were then washed and incubated with custom Stellaris probes in FISH buffer (10% formamide, 1X SSC, 1 mg/ml yeast tRNA, 2 mM Vanadyl Ribonucleoside complex) at 30°C overnight. Cells were then washed in SSC + 10% formamide prepared fresh twice for 30 min at 30° C. Cells were mounted in Vectashield and imaged with a Cy5 filter (Zeiss 50).

Data were analyzed by first removing background fluorescence. Cells were manually outlined and GFP signal was segmented from a maximum intensity projection using Otsu thresholding following subtraction of background signal in FIJI. The most in-focus slice was identified. Next, mRNAs were

identified using the Find Maxima tool and each was outlined. mRNAs in the same slice as the G body or one slice away were recorded. The Euclidean distance between the G body center and each point in each of the mRNAs within the same plane as the G body was measured with Python scripts. Distance distributions were pooled for all experiments.

## FRAP acquisition and analysis

Cells were reinocculated from stationary cultures into the indicated media and grown for 18 hr in hypoxia. Cells were imaged on a Nikon Ti Eclipse microscope with a 100X objective using a FITC filter. After a Z stack was captured, individual pixels were chosen to bleach. G bodies were bleached with a 50 mW 405 nm laser at 40% laser power for 50 ms. Z stacks were then captured every minute for 40 min. Background signal was subtracted and data were normalized for photobleaching from the average of three unbleached control cells in each image. G bodies were tracked by measuring a circle around the brightest focus in each cell with a radius determined by maximizing the percent of signal bleached. Only traces with <40% of fluorescence remaining in the G body were considered. Data were then normalized from 0 (post-bleach) to 1 (pre-bleach) and averaged across replicates. Recovery rates were determined by fitting to an exponential decay function:

$$I(t) = A\left(1 - e^{-\tau t}\right)$$

where I is the normalized intensity, A is the amplitude of recovery, the recovery half time is $\ln(0.5)/\tau$, and t is the time in minutes. The half time and extent of recovery were reported. Experiments with plasmids suffered from substantial photobleaching leading to unreliable results after 25 min and thus were fit only for the first 25 min of recovery.

## Growth competition assay

Cells carrying plasmids that prevent G body formation (GB–) in the presence of $CuSO_4$ and those carrying plasmids that do not prevent G body formation (GB+) were reinocculated from overnight cultures in SMD(-Ura) and grown to log phase in SMD(-Ura). Cells were then mixed in SMD(-Ura) + 50 µM $CuSO_4$ either singly at an $OD_{600}$ of 0.05 or together at an $OD_{600}$ of 0.025 each. Cells were grown in hypoxia for 20 hr. For multiple generation assays, serial dilutions of cultures were made and imaged after 24 and 48 hr in hypoxia. ODs were measured and cells were imaged. The fraction of cells from the GB+ and GB– cells in the mixed culture was calculated as:

$$f_{GB+} = \frac{\%_{mix} - \%_{GB-}}{\%_{GB+} - \%_{GB-}}$$

where $\%_{mix}$ is the % of cells from the mixed cultures with G bodies, $\%_{GB+}$ is the % of cells from the GB+ strain alone with G bodies and %GB– is the percent of cells from the GB– strain alone with G bodies. Relative proportions of cells at 0 generations were determined by spotting 10 µl of a $1:10^3$ dilution of each population at $OD_{600} = 0.1$ to YPD and counting colony forming units (CFU).

## Acknowledgements

We thank members of the Kim Lab (Amelia Alessi, Charlotte Choi, Mindy Clark, Jessica Kirshner, Alex Rittenhouse, Margaret Starostik, and Rebecca Tay) and Tatjana Trcek for helpful suggestions. We also thank Himani Galagali and Angela Anderson for comments on the manuscript. This work was supported by a grant from the NIH (R01GM129301) and a grant from the National Institute of Neurological Disorders and Stroke, National Alzheimer's Project Act (NAPA) 1-RF1-NS113636-01.

# Additional information

## Funding

| Funder | Grant reference number | Author |
|---|---|---|
| National Institute of General Medical Sciences | RO1GM129301 | Gregory Fuller<br>Ting Han<br>Mallory A Freeberg<br>John K Kim |
| National Institute of General Medical Sciences | P41 GM103533 | James J Moresco<br>John R Yates III III |
| National Institute of Neurological Disorders and Stroke | 1-RF1-NS113636-01 | Amirhossein Ghanbari Niaki<br>Sua Myong |

The funders had no role in study design, data collection and interpretation, or the decision to submit the work for publication.

## Author contributions

Gregory G Fuller, Conceptualization, Resources, Data curation, Formal analysis, Investigation, Visualization, Methodology; Ting Han, Conceptualization, Investigation, Methodology; Mallory A Freeberg, Data curation, Software, Formal analysis, Investigation, Visualization; James J Moresco, Data curation, Formal analysis, Investigation; Amirhossein Ghanbari Niaki, Nathan P Roach, Sua Myong, Methodology; John R Yates III, Resources, Supervision, Funding acquisition; John K Kim, Conceptualization, Supervision, Funding acquisition, Methodology, Project administration

## Author ORCIDs

Gregory G Fuller (ID) https://orcid.org/0000-0003-2960-0256
John R Yates III (ID) https://orcid.org/0000-0001-5267-1672
John K Kim (ID) https://orcid.org/0000-0001-9838-3254

## Decision letter and Author response

Decision letter https://doi.org/10.7554/eLife.48480.sa1
Author response https://doi.org/10.7554/eLife.48480.sa2

# Additional files

## Supplementary files

• Supplementary file 1. Mass spectrometry identification of mRNA binding proteins. Related to *Figure 1*.

• Supplementary file 2. qPCR primers used. Related to *Figure 2—figure supplement 1*.

• Supplementary file 3. Strains used in this study.

• Supplementary file 4. Results of G body RNA sequencing. Related to *Figures 2* and *3*.

• Supplementary file 5. Source code used to analyze G body size in *Figure 7—figure supplement 1A*.

• Transparent reporting form

## Data availability

Sequencing data have been deposited in GEO under accession code GSE65992 and GSE145881. Summaries of these data are available in Supplementary file 4. Mass spectrometry data are available in Supplementary file 1. Source data has been provided for graphs in Figures 2, 3, 4, 5, 6, 7 and supplementary figures.

The following datasets were generated:

| | Database and |
|---|---|

| Author(s) | Year | Dataset title | Dataset URL | Identifier |
|-----------|------|---------------|-------------|------------|
| Freeberg MA, Han T, Kim JK | 2017 | Glycolytic Enzymes Bind mRNA and Coalesce in G-bodies Essential for Cell Proliferation Under Hypoxic Conditions | https://www.ncbi.nlm.nih.gov/geo/query/acc.cgi?acc=GSE65992 | NCBI Gene Expression Omnibus, GSE65992 |
| Fuller GF, Kim JK | 2020 | Glycolytic body (G body) RIP-sequencing | https://www.ncbi.nlm.nih.gov/geo/query/acc.cgi?acc=GSE145881 | NCBI Gene Expression Omnibus, GSE145881 |

The following previously published datasets were used:

| Author(s) | Year | Dataset title | Dataset URL | Database and Identifier |
|-----------|------|---------------|-------------|-------------------------|
| Freeberg MA, Han T, Moresco JJ, Kong A, Yang Y, Lu Z, Yates JR, Kim JK | 2013 | Pervasive and dynamic protein binding sites of the mRNA transcriptome in Saccharomyces cerevisiae | https://www.ncbi.nlm.nih.gov/geo/query/acc.cgi?acc=GSE43747 | NCBI Gene Expression Omnibus, GSE43747 |

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
