## [Decision Letter]

**Acceptance summary:**

This paper implicates RNA and the ability to bind RNA in the formation of assemblies of glycolytic enzymes, so called G-bodies observed in yeast under hypoxic conditions. Though the precise chain of causality, linking hypoxia-induced changes in RNA binding properties of the glycolytic enzymes, to the dynamics of G-bodies remain unresolved, the evidence for RNA's role in the process is deemed of sufficient value to merit publication in *eLife*.

**Decision letter after peer review:**

Thank you for submitting your article "RNA promotes phase separation of glycolysis enzymes into yeast G bodies in hypoxia" for consideration by *eLife*. Your article has been reviewed by two peer reviewers, and the evaluation has been overseen by David Ron as the Senior and Reviewing Editor. The reviewers have opted to remain anonymous.

The reviewers have discussed the reviews with one another and the Editor has drafted this decision to help you prepare a revised submission.

Summary:

This manuscript builds on a previous work establishing that glycolytic enzymes coalesce in G bodies under hypoxic stress conditions in yeast. The authors describe for the first time the presence of RNA in yeast G bodies, membraneless cytoplasmic granules where the glycolytic machinery concentrates under hypoxic stress conditions, and explore the role that RNA may play in the formation/maintenance of these structures. The main message of the manuscript is that RNA binding conveys specific properties to G bodies. This message, if adequately supported by the experimental data, is deemed sufficiently important to merit publication in *eLife*. However, the expert reviewers (whose comments follow, unedited) have identified several limitations to the work, as it currently stands, limitations that undermine the strength of the conclusions drawn.

Essential revisions:

1) The role of the RNA component in supporting the key features of the G-bodies: namely their ability to fuse and display properties that makes them similar to stress granules, needs to be strengthened. This after all is the main conclusion of the study.

2) The key features of the glycolytic enzymes as RNA binding proteins have been established experimentally under normoxic conditions, yet the premise of the paper is that such interactions are required for formation of G-bodies under hypoxic conditions. This limitation needs to be addressed.

3) The functional role of the association between enzymes and RNAs as defined by the discovery of binding sites on the various mRNAs, needs to be addressed.

Reviewer #1:

In the manuscript "RNA promotes phase separation of glycolysis enzymes into yeast G bodies in hypoxia", Gregory G. Fuller and co-authors describe for the first time the presence of RNA in yeast G bodies, membraneless cytoplasmic granules where the glycolytic machinery concentrates under hypoxic stress conditions, and explore the role that RNA may play in the formation/maintenance of these structures. The starting point of this study is the discovery of the yeast mRNA-bound proteome, where most of the glycolysis enzymes were identified as RNA binding proteins. Indeed, the capacity of many of these enzymes to bind RNA had been described in earlier studies. The selective degradation of RNAs at G bodies affects the integrity/dynamics of these structures, demonstrating that RNA could convey at least some of the phase separation properties of these G bodies which are somewhat similar to stress granules (but not P bodies).

As a nice follow up of the important previous work from Dr. Kim's group, this work brings new insights into the structure and regulation of G bodies, which is of course of great interest for the RNA and cell biology communities.

The main message of the manuscript is that RNA binding conveys specific properties to G bodies. However, the experimental path followed is sometimes not straight enough, or insufficient to support the author's claims. These are some of the main points to address:

1) The authors establish the capacity of glycolysis enzymes to bind RNA under normoxic conditions, but not under hypoxic stress when G bodies are assembled. Could stress modulate the capacity of G body components to bind RNA?

2) Similarly, the RNA targets of some of the main G body components are only defined under normoxic conditions. Thus, the fact that glycolysis enzymes bind to specific sites in the yeast transcriptome under non stress conditions may not be related to the formation or dynamics of G bodies.

3) Again, the role that the association between glycolysis enzymes and their target mRNAs is not addressed in a straightforward manner. For instance, the authors identify putative elements to which Eno1/Eno2/Pfk2/Fba1 may bind, but then they do not demonstrated whether these binding sites serve to target mRNAs into G bodies. Are these mRNAs translated in the vicinity/surface of G bodies? Are G bodies concentrating the majority of these mRNAs in foci, or just a small fraction?

4) The authors demonstrate that G bodies can fuse and display properties that makes them similar to stress granules. While it is tempting to speculate that these properties are at least in part due to the RNA component of G bodies, this role is not established in a direct, convincing manner. Alongside this observation, it may be convenient to consider if the experimental evidence meets the main statement of the work represented in its title.

These caveats should be addressed properly. Nevertheless, it is important to remark the novelty and relevance of this work, and the overall quality of the experimental evidence presented.

Reviewer #2:

This manuscript builds on a previous work establishing that glycolytic enzymes coalesce in G bodies under hypoxic stress conditions in yeast, and it aims to examine the physical properties and properties of G body formation. The authors elegantly demonstrate that glycolysis enzymes can bind a specific set of mRNA that are required for G body formation, maintenance and structural integrity. These structures are suggested to be formed via phase separation mechanism in vivo. Finally, authors suggest a model in which G bodies enhance the rate of glycolysis when cellular demand for energy rely on it. In this respect, it would be effective to show that disruption of G bodies in hypoxic conditions, by targeting non-specific RNAse impact glycolysis output (for example using the fluorometric assay presented in Jin et al., 2017) or cell survival. Although not required for the main message of the work (e.g. that RNA promotes phase separation of glycolysis enzymes into yeast G bodies in hypoxia) but it will enhance the biological relevance of RNA-mediated G bodies formation. Overall, the work is carefully done.

[Editors' note: further revisions were suggested prior to acceptance, as described below.]

Thank you for re-submitting your article "RNA promotes phase separation of glycolysis enzymes into yeast G bodies in hypoxia" for consideration by *eLife*. Your article has been reviewed by two peer reviewers, and the evaluation has been overseen by David Ron as the Senior and Reviewing Editor. The reviewers have opted to remain anonymous.

Summary:

This a revised version of a paper initially reviewed in August 2019. As noted then, the authors describe for the first time the presence of RNA in yeast G bodies, membraneless cytoplasmic granules where the glycolytic machinery concentrates under hypoxic stress conditions, and explore the role that RNA may play in the formation/maintenance of these structures. The main message of the manuscript is that RNA binding conveys specific properties to G bodies.

Certain limitation were noted at the time and these have been largely addressed in the revised version. While the manuscript has been improved, two important issues remain.

Essential revisions:

The chain of causality implicating mRNA binding by glycolytic enzymes in the hypoxia-dependent formation of G-bodies may in fact be inverted. This is not detrimental to your conclusions – mRNAs are still drivers of G-bodies – but it raises the possibility that the detailed mechanism you propose may not be correct.

We also note a limitation in the interpretation of the effect of targeting nucleases to the G-bodies, as such crude tools may have all sorts of pleiotropic consequences.

Ordinarily, we would have set as conditions for acceptance of the manuscript that you address point 1 of reviewer #1's comments experimentally; namely that you devise experiment to compare the association of glycolytic enzymes with their target RNAs under normoxic and hypoxic conditions, and that you address editorially the limitations in interpreting the consequences of unleashing upon the cells an endonuclease – even one ostensibly localised to a specific site.

However, given the current pandemic, we appreciate that it may be a while before any experiment gets done anywhere. Therefore we would like to offer you the choice of (a) responding editorially to both critiques and we will consider a revised manuscript along these lines or (b) addressing the issue experimentally, and *eLife* shall provide you with an effectively indefinite extension to do so.

Reviewer #1:

The updated version of Gregory Fuller et al. has been improved and is now a better, more complete contribution. As a general criticism, the experimental evidence is in agreement with the main conclusions of this work, but in many instances it does not provide sufficient support to the authors' model.

1) smFISH and G body purification experiments show that only a minor fraction of G body component-interacting mRNAs are recruited to G bodies under hypoxia. Most likely, the recognition of specific cis acting elements by glycolytic enzymes should not serve to regulate the expression of target mRNAs at G bodies under hypoxic conditions. Instead, the authors propose that RNA molecules work as a co-factor for the assembly/maintenance of G bodies.

Alternatively, the well-characterized binding of glycolytic enzymes to specific 3'UTR elements under normoxic conditions could be relevant precisely under normoxic conditions, where these enzymes would contribute to regulate mRNA translation or stability of their target mRNAs. Hypoxia may alter these interactions, such that the bulk of glycolytic enzyme-bound mRNAs would be released from these mRNAs and concentrate in G bodies (in a process that could still be facilitated by RNA). To distinguish between these two hypotheses, it would have been interesting to address if the association between glycolytic enzymes and RNA is quantitatively altered by hypoxic conditions.

Unfortunately, the methodological approaches conducted to examine the association of glycolytic enzymes to RNA in normoxic and hypoxic conditions are different (crosslinking and immunoprecipitation in normoxic conditions, G body purification and smFISH under hypoxia), and for this reason it is hard to tell if hypoxia enhances or reduces the association of glycolytic enzymes to RNA. While the model proposed in this manuscript is plausible and attractive, the authors should either test directly if hypoxic conditions reduce/increase RNA binding capacity of G body components or, at least, mention alternative explanations to their observations in the Discussion.

2) Many of the conclusions of this work are supported by the artificial targeting of RNAses to G bodies, which is achieved by the fusion of Pfk2 to RNAse A or Mqs-R, expressed from the copper-inducible, CUP1 promoter. This tool is used to establish the functional importance of RNA in the assembly and function of G bodies, as well as in the dynamic properties of these foci. Since these experiments provide essential support for the model proposed, the authors should evaluate if the main effects derived from the expression of these RNAse fusion proteins result from the disruption of G bodies, or from the degradation of mRNAs that bound by Pfk2. In other words, they should test if these RNAse fusion proteins reduce the levels of G body components (or, in a broader sense, the levels of mRNAs bound by Pfk2), which could affect cell fitness and G body integrity in a way that is not considered in the current manuscript.

Reviewer #2:

The authors have responded to all the reviewers' critiques either by additional experiments, including, Figure 3, Figure 7 and Figure 2—figure supplement 1D, Figure 3—figure supplement 1, Figure 7—figure supplements 2 and 4), or by modulating the text in the Discussion. Overall, the work is carefully done and I have no more comments. I recommend the revised manuscript for publication in *eLife*.

---

## [Author Response]

Essential revisions:1) The role of the RNA component in supporting the key features of the G-bodies: namely their ability to fuse and display properties that makes them similar to stress granules, needs to be strengthened. This after all is the main conclusion of the study.

We agree that it is important to demonstrate a relationship between RNA binding and the physical properties of G bodies. Several lines of evidence suggest that RNA is important for G body physical properties. First, we demonstrate that RNA is required for G body formation by targeting RNases to nascent G bodies. Second, when RNase is directed to G bodies that already exist, we observe multiple foci. If these foci, like nascent G bodies, can fuse, then we would observe large single foci. However, this is not the case. This point is now highlighted in the Discussion. Third, we have now performed FRAP experiments in vivo. In cells in which we have weakly induced Pfk2-MqsR-Flag, we observe slightly greater recovery rates than cells with a vector control suggesting that loss of RNA enhances Pfk2-GFP dynamics in G bodies (Figure 7—figure supplement 2D). This is consistent with weaker association of Pfk2-GFP in G bodies as it is now more able to exchange with the cytoplasmic fraction. Thus, RNA may promote fusion by stabilizing binding of granule components.

2) The key features of the glycolytic enzymes as RNA binding proteins have been established experimentally under normoxic conditions, yet the premise of the paper is that such interactions are required for formation of G-bodies under hypoxic conditions. This limitation needs to be addressed.

We agree that identifying the RNAs in G bodies in hypoxia is important. We have done substantial additional experiments to identify and validate these RNA interactions with G bodies. We have applied two orthogonal methods of analyzing G body localization of RNA. In particular, we have sequenced RNAs from purified G bodies as well as total RNA and flow through RNA to identify transcripts enriched in G bodies (Figure 3A, B, Figure 2—figure supplement 1D, Supplementary file 4). The overlap of RNAs bound by glycolysis enzymes in normoxia and the most enriched RNAs in purified G bodies is significantly nearly 2-fold greater than expected by chance (Figure 3C, D). We have validated that RNAs localize to G bodies by smFISH with varying levels of G body enrichment (Figure 3E). These two methods agree in that the most enriched RNAs in G body RIP-seq are also the most enriched by smFISH (Figure 3F, G). However, surprisingly, while some RNAs are enriched in G bodies, the majority of any given RNA species is localized outside of G bodies. Thus, while there is some degree of specificity in the RNAs recruited to G bodies, G bodies are not highly concentrating particular RNAs.

3) The functional role of the association between enzymes and RNAs as defined by the discovery of binding sites on the various mRNAs, needs to be addressed.

The observation that RNAs bound to glycolysis enzymes in normoxia are overrepresented in G body associated RNAs (Figure 3C, D) suggests that binding of glycolysis enzymes to RNA is important for the localization of RNAs to G bodies. However, only a small fraction of even the most enriched mRNAs localizes to G bodies (Figure 3F). Thus, G bodies likely have a limited, if any, role in regulating the translation or stability of constituent RNAs. Thus, while binding of RNA to G body proteins is important for the recruitment of each to G bodies, the formation of G bodies appears to be the function of this association. To test this point, we used a growth competition assay to determine the relative fitness of cells with and without G bodies (Figure 7—figure supplement 4). Cells able to form G bodies consistently outcompeted cells unable to form G bodies, suggesting that RNA nucleation of G bodies has a functional output in cells.

Reviewer #1:[…] The main message of the manuscript is that RNA binding conveys specific properties to G bodies. However, the experimental path followed is sometimes not straight enough, or insufficient to support the author's claims. These are some of the main points to address:1) The authors establish the capacity of glycolysis enzymes to bind RNA under normoxic conditions, but not under hypoxic stress when G bodies are assembled. Could stress modulate the capacity of G body components to bind RNA?

We agree that this was a weakness of our earlier submission. We have addressed this concern through RNA sequencing of G body-associated RNAs and smFISH in hypoxia cells (see also response to Essential Revision #2, Figure 3, and Figure 3—figure supplement 1).

2) Similarly, the RNA targets of some of the main G body components are only defined under normoxic conditions. Thus, the fact that glycolysis enzymes bind to specific sites in the yeast transcriptome under non stress conditions may not be related to the formation or dynamics of G bodies.

In our analysis of G body RNA sequencing data, we identify significant overlap between those RNAs bound by glycolysis enzymes and the most enriched RNAs in G bodies suggesting that this normoxic RNA binding is relevant to RNA recruitment to G bodies (Figure 3C, D).

3) Again, the role that the association between glycolysis enzymes and their target mRNAs is not addressed in a straightforward manner. For instance, the authors identify putative elements to which Eno1/Eno2/Pfk2/Fba1 may bind, but then they do not demonstrated whether these binding sites serve to target mRNAs into G bodies. Are these mRNAs translated in the vicinity/surface of G bodies? Are G bodies concentrating the majority of these mRNAs in foci, or just a small fraction?

Our findings indicate that many of the RNAs bound by glycolysis enzymes in normoxia bind to are present in G bodies. However, there are many other RNAs as well amounting to hundreds of RNAs in total (Figure 3). Furthermore, using smFISH, we found that even the most enriched G body transcripts are only modestly concentrated in G bodies and that only a small fraction of any mRNA will localize to G bodies (Figure 3F, G). Taken together with our nuclease treatments demonstrating the necessity of RNA for G body formation, these observations suggest that while there must be RNA present for G body formation, the identity of the RNA may not be important for G body formation. The binding sites of the glycolysis enzymes on various mRNAs, then, influence the likelihood of that RNA being recruited to G bodies and allow these mRNAs to seed G body formation. It is unlikely, however, given the small fraction of any mRNA in G bodies, that G body localization plays a substantial role in affecting overall translation rates, and any local translation near G bodies will therefore be difficult to determine. Future studies may be able to address this possibility and determine if there are specific mRNAs required for G body formation.

4) The authors demonstrate that G bodies can fuse and display properties that makes them similar to stress granules. While it is tempting to speculate that these properties are at least in part due to the RNA component of G bodies, this role is not established in a direct, convincing manner. Along this this observation, it may be convenient to consider if the experimental evidence meets the main statement of the work represented in its title.

Please see our response to Essential Revision #3 above.

Reviewer #2:This manuscript builds on a previous work establishing that glycolytic enzymes coalesce in G bodies under hypoxic stress conditions in yeast, and it aims to examine the physical properties and properties of G body formation. The authors elegantly demonstrate that glycolysis enzymes can bind a specific set of mRNA that are required for G body formation, maintenance and structural integrity. These structures are suggested to be formed via phase separation mechanism in vivo. Finally, authors suggest a model in which G bodies enhance the rate of glycolysis when cellular demand for energy rely on it. In this respect, it would be effective to show that disruption of G bodies in hypoxic conditions, by targeting non-specific RNAse impact glycolysis output (for example using the fluorometric assay presented in Jin et al., 2017) or cell survival. Although not required for the main message of the work (e.g. that RNA promotes phase separation of glycolysis enzymes into yeast G bodies in hypoxia) but it will enhance the biological relevance of RNA-mediated G bodies formation. Overall, the work is carefully done.

Degrading G bodies using Pfk2-MqsR-Flag and Pfk2-RNase A requires us to express each protein in addition to the endogenous protein. Thus, this may influence glucose flux rates. However, we have included a growth competition assay that demonstrates a distinct advantage for cells that are able to form G bodies relative cells that cannot form G bodies.

[Editors' note: further revisions were suggested prior to acceptance, as described below.]

Essential revisions:The chain of causality implicating mRNA binding by glycolytic enzymes in the hypoxia-dependent formation of G-bodies may in fact be inverted. This is not detrimental to your conclusions – mRNAs are still drivers of G-bodies – but it raises the possibility that the detailed mechanism you propose may not be correct.

We thank reviewer 1 for bringing up this issue and have now added text to further clarify our conclusions in the consider the alternative model in the Discussion section:

“Alternatively, hypoxia may weaken interactions of glycolysis enzymes with RNAs allowing for G bodies to form via protein-protein interactions. Many RNAs are localized peripherally to G bodies (Figure 3). These RNAs may then allow G bodies to fuse by binding the glycolysis enzymes in other G bodies, explaining the loss of foci with loss of RNA. Experiments measuring binding affinity of glycolysis enzymes to mRNA in hypoxia and normoxia will be required to differentiate these possibilities.”

We also note a limitation in the interpretation of the effect of targeting nucleases to the G-bodies, as such crude tools may have all sorts of pleiotropic consequences.

We have added text to the results to clarify the logic of our interpretation as well as acknowledge this caveat in the Results section:

“Taken together, these data indicate that 1) Nascent G body formation was inhibited by RNases in a concentration-dependent manner (Figure 4A); 2) Inhibition of de novo G body formation required both RNase activity and targeting to the site of G body formation (Figure 4C-E); 3) Once formed, targeting an RNase to G bodies led to cells with multiple foci, suggesting fragmentation but not dissolution of G bodies (Figure 5D-F); and 4) Once formed in hypoxia, G bodies could persist for tens of hours in normoxia (Figure 5D-F). […] Loss of G body components is unlikely to explain the fracturing of extant G bodies following induction after shifting from hypoxia to normoxia.”